# Interactions between lineage-associated transcription factors govern haematopoietic progenitor states

Iwo Kucinski[1] [ID], Nicola K Wilson[1], Rebecca Hannah[1], Sarah J Kinston[1], Pierre Cauchy[2], Aurelie Lenaerts[2,3], Rudolf Grosschedl[2] [ID] & Berthold Göttgens[1,*] [ID]

## Abstract

Recent advances in molecular profiling provide descriptive data-sets of complex differentiation landscapes including the haematopoietic system, but the molecular mechanisms defining progenitor states and lineage choice remain ill-defined. Here, we employed a cellular model of murine multipotent haematopoietic progenitors (Hoxb8-FL) to knock out 39 transcription factors (TFs) followed by RNA-Seq analysis, to functionally define a regulatory network of 16,992 regulator/target gene links. Focussed analysis of the subnetworks regulated by the B-lymphoid TF Ebf1 and T-lymphoid TF Gata3 revealed a surprising role in common activation of an early myeloid programme. Moreover, Gata3-mediated repression of Pax5 emerges as a mechanism to prevent precocious B-lymphoid differentiation, while Hox-mediated activation of Meis1 suppresses myeloid differentiation. To aid interpretation of large transcriptomics datasets, we also report a new method that visualises likely transitions that a progenitor will undergo following regulatory network perturbations. Taken together, this study reveals how molecular network wiring helps to establish a multipotent progenitor state, with experimental approaches and analysis tools applicable to dissecting a broad range of both normal and perturbed cellular differentiation landscapes.

**Keywords** haematopoiesis; network; progenitors; scRNA-Seq; transcription factor

**Subject Categories** Chromatin, Transcription & Genomics; Development; Haematology

**The EMBO Journal (2020) 39: e104983**

## Introduction

Mature blood cells are continuously replenished by a flow of differentiating cells originating from multipotent, self-renewing haematopoietic stem cells (HSCs), which give rise to multi, oligo and unipotent progenitors with decreasing self-renewal potentials. Potential structures for this differentiation hierarchy ("haematopoietic tree") have been proposed through decades of iterative sampling of cell subpopulations and functional testing using transplantation or colony assays (Eaves, 2015; Laurenti & Göttgens, 2018). More recently, single-cell functional assays, scRNA-Seq and barcoding approaches emphasise the landscape view of haematopoietic differentiation, proposing more gradual differentiation trajectories and a more probabilistic nature of lineage choices (Fig 1A; Nestorowa et al, 2016; Pei et al, 2017; Dahlin et al, 2018; Rodriguez-Fraticelli et al, 2018; Tusi et al, 2018; Watcham et al, 2019; Weinreb et al, 2020). Importantly, models based merely on the cataloguing of molecular data remain descriptive, with little insight into the mechanisms behind cellular decision making as cells traverse the differentiation landscape. The concept of differentiation landscapes was introduced by Waddington (Waddington, 1957), who proposed, right from the start, that beneath the landscape there had to be a complex molecular network which by, determining the shape of the landscape, controls cellular decision making.

Deciphering complex regulatory networks constitutes a formidable task due to the large number of components and an even larger number of possible interactions. For the past decade or so, much hope has been pinned on inference based on correlative evidence, e.g. trying to explain transcriptional regulation from variation in gene expression across many conditions or samples. Correlative inference approaches, however, commonly lack the means to identify causality or directionality. Regression-based methods, bayesian networks and differential equation models have all been proposed to overcome these shortcomings (Sanguinetti & Huynh-Thu, 2019), but unfortunately have had limited success so far (Marbach et al, 2012; Chen & Mar, 2018; Pratapa et al, 2020). Genome-wide binding profiles are often used for cross-validation. These, however, also face limitations, because singular TF binding events constitute a poor predictor of gene regulation (ENCODE Project Consortium, 2012; Calero-Nieto et al, 2014; Kellis et al, 2014; Vijayabaskar et al, 2019). Arguably, the main limitation is

1 Wellcome–MRC Cambridge Stem Cell Institute, Department of Haematology, Jeffrey Cheah Biomedical Centre, University of Cambridge, Cambridge, UK
2 Department of Cellular and Molecular Immunology, Max Planck Institute of Immunobiology and Epigenetics, Freiburg, Germany
3 International Max Planck Research School for Molecular and Cellular Biology, Max Planck Institute of Immunobiology and Epigenetics, Freiburg, Germany
  *Corresponding author. Tel: +44 1223 336829; E-mail: bg200@cam.ac.uk

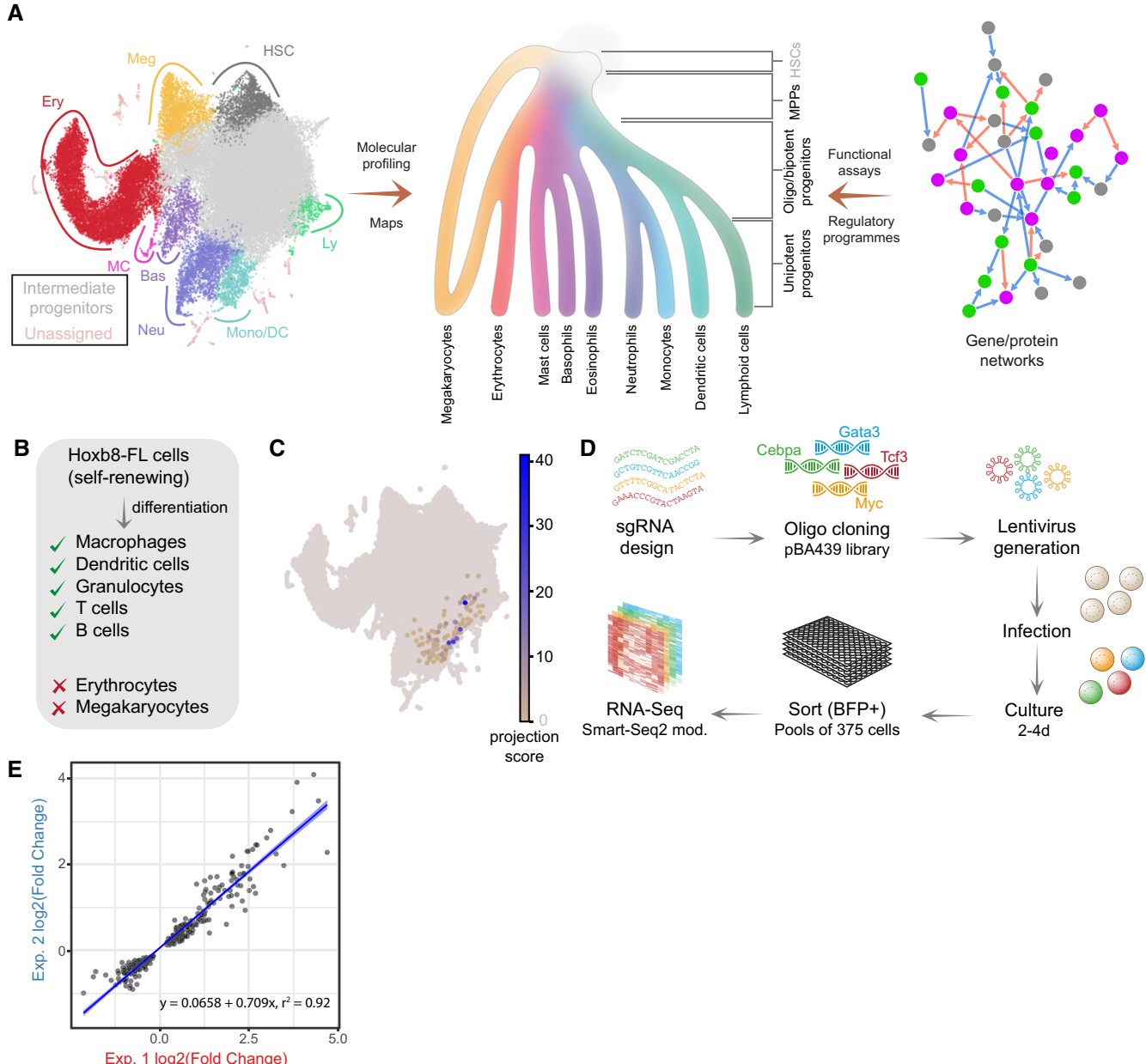

**Figure 1. CRISPR/Cas9 screen with a transcriptomics readout.**

A   Understanding the cell state landscapes of haematopoietic progenitors. (left) Annotated UMAP projection of a scRNA-Seq landscape—mouse LK + LSK populations (Dahlin *et al*, 2018) (middle) diagram representing haematopoietic hierarchy with gradual changes in cell fate potential (colour gradient) from HSCs to differentiated states (adapted from the Molecular Cell Biology, 9th edition *under preparation*). (right) Diagram of a molecular network.

B   Differentiation capacity of Hoxb8-FL cells.

C   Projection of Hoxb8-FL transcriptome onto the LK/LSK mouse landscape. The projection score (based on nearest neighbours) reflects relative transcriptional similarity to the Hoxb8-FL state for each LK/LSK cell. For the majority of cells, no neighbours are identified (grey), some cells exhibit low similarity (yellow), and a small set of cells exhibit high similarity (blue).

D   Schematic of the screen layout, sgRNAs were cloned into the pBA439 backbone and introduced into Hoxb8-FL cells via lentiviral infection. Cells were cultured for either 2 or 4 days, followed by sorting for cells carrying sgRNA constructs (BFP+) into small pools and subsequent small-scale RNA-Seq analysis.

E   Reproducibility of 2 independent experiments—correlation of observed changes in expression. Blue line indicates the linear fit with shaded areas as confidence intervals.

Data information: Abbreviations: Meg—megakaryocytes, HSC—haematopoietic stem cells, Ery—erythrocytes, MC—mast cells, Bas—basophils, Mono—monocytes, DC—dendritic cells, Neu—Neutrophils, Ly—lymphoid, Myo—myeloid.

the lack of gold standards—sets of verified, functional connections which can be used to objectively evaluate and refine inference methods.

There is a growing appreciation therefore that renewed emphasis needs to be given to direct experimental intervention as the way of identifying causal links. Targeted genetic/chemical perturbations

have been used to reconstruct small networks (Jaeger, 2011; Briscoe & Small, 2015; Hill *et al*, 2016) with considerable success. However, scalability of conventional "functional" experiments has been limited. The CRISPR/Cas9 revolution has now firmly established the feasibility of large-scale gene perturbation screens. Moreover, the miniaturisation of next-generation sequencing protocols allows for significant cost savings thus enabling scalable genetic perturbations with simultaneous transcriptomic readout (Datlinger *et al*, 2017).

In this study, we constructed an experimentally defined network connecting 39 TFs—chosen key regulators of haematopoietic differentiation—with their downstream targets. Due to extensive heterogeneity of primary cells and difficulties in maintaining their steady-state *ex vivo*, we utilised a multipotent cell line model—Hoxb8-FL (Redecke *et al*, 2013). By establishing a scalable screening pipeline to knock out single TFs and analyse the resulting transcriptomic changes by RNA-Seq, we identified 16,992 TF-target regulatory links across 7,388 target genes, revealing a range of target gene modules associated with specific functions including the maintenance of self-renewal and preventing dominance of specific lineage-specific programmes. To help attribute biological functions to analysed TFs, we also propose a new method—DoT score—which aids interpretation of transcriptomic changes using scRNA-Seq landscapes as a reference.

# Results

### A sensitive and scalable method to infer TF-target connections

Hoxb8-FL cells represent a functional *in vitro* counterpart to lymphoid-primed multipotent progenitors (LMPP), which can be maintained as a self-renewing culture in the presence of Flt3 ligand and activation of a Hoxb8 oestrogen receptor fusion transgene, and can differentiate to myeloid and lymphoid cells both *in vitro* and *in vivo* (Redecke *et al*, 2013) (Fig 1B). To relate Hoxb8-FL cells to their likely counterparts in primary cell transcriptional landscapes, we identified the nearest neighbour cells connecting our previously published landscape of over 40,000 mouse HSPCs (Dahlin *et al*, 2018) with 82 single-cell transcriptomes from Hoxb8-FL cells cultured in self-renewal conditions (Basilico *et al*, 2020). As shown in Fig 1C, the primary HSPC cells that are most transcriptionally similar to Hoxb8-FL cells occupy a defined territory between myeloid and lymphoid progenitors, consistent with their LMPP-like properties. The value of Hoxb8-FL cells as a model for haematopoietic progenitors is enhanced further by previously generated genome-wide CRISPR/Cas9-dropout screen data (Basilico *et al*, 2020), which highlight genes critical for self-renewal of Hoxb8-FL cells.

To establish functional links between TFs and their targets, we developed a CRISPR/Cas9-RNA-Seq screening approach (Fig 1D). Each TF was perturbed independently by three sgRNAs, introduced via lentiviral infection into Cas9-expressing Hoxb8-FL cells. These were subsequently analysed for transcriptomic changes after 2 or 4 days using an adapted Smart-Seq2 protocol (Picelli *et al*, 2014; Bagnoli *et al*, 2018) on 8 pools each of 375 cells. We avoided previously reported barcode recombination (Xie *et al*, 2018) by producing viral particles and infecting cells separately. As a negative control, we used two control constructs: sgRNA targeting GFP

(sequence not present in the genome) and sgRNA targeting the *Rosa26* locus. In parallel, we analysed pools of cells after switching off Hoxb8 ectopic expression for 18 h but maintaining Flt3L signalling (Hoxb8*), a condition ultimately leading to dendritic cell differentiation after 4–5 days.

Gene knockout efficiency was confirmed by targeting the ubiquitously expressed CD45 locus, which was successfully inactivated in 48% of cells (Fig EV1A). Moreover, CRISPR/Cas9 perturbation also resulted in the loss of the corresponding TF protein as validated by the absence of Gata3 ChIP-Seq signal in single-cell clones derived from cells targeted with the Gata3 guide RNAs (Appendix Fig S6). Furthermore, high-throughput sequencing of loci targeted by 11 sgRNAs across 4 genes showed consistent frameshift in 30–50% DNA copies (Fig EV1B, Table EV1), indicating that targeted populations will contain some heterozygous and WT cells despite efficient editing. To ensure high-sensitivity in detecting expression changes, we therefore performed 8 replicate RNA-Seq experiments per condition (Fig EV1C). Differential expression (DE) statistic between matching perturbed and control samples was used to identify regulator–target relationships, with the observed $log_2$(fold change) providing the weights for the resulting network edges. Two independent experiments targeting Gata3 show strong overlap and effect correlation across target genes (Fig 1E), and there is a strong correlation among the 3 sgRNAs targeting the same gene (Fig EV1D and F).

Choice of time-point for the analysis is critical. There is a fine balance between the risk of analysing cells before the protein is sufficiently depleted if analysed too early and skewing data towards secondary (and higher order) effects at later time-points. Additionally, it takes approximately 1 day for the viral construct to integrate and transcribe/translate after the infection. When targeting the non-essential Gata3, we observed robust and reproducible signal between days 3 and 5 after perturbation (Fig EV1E); hence, we chose the 4 day time-point to provide sufficient time for gene knockout effects. For essential genes, we analysed cells mostly after 2 days to precede the drop in cell survival, as observed after removing Cebpa or Myc (Fig EV1G).

### A functional network of haematopoietic transcription factors

We next applied the approach outlined above to identify the downstream targets of 38 TFs, chosen based on their haematopoietic function and expression in progenitor cells (Dataset EV1). We also assayed a cohesin complex component—Rad21, which plays an important role in haematopoiesis (Panigrahi & Pati, 2012) and regulates expression of pluripotency genes (Nitzsche *et al*, 2011). Twelve out of these 39 genes are essential for survival of Hoxb8-FL cells, i.e. their knockout leads to a competitive disadvantage when cultured with WT cells (Basilico *et al*, 2020) ("Dropout TFs"). Bioinformatic analysis of the more than 1,000 newly generated RNA-Seq datasets revealed a network of 39 TFs connected via 16,992 edges with 7,388 downstream target genes, i.e. differentially expressed following perturbation of one or more TFs (Dataset EV2). The number of differentially regulated genes included within the network is dependent on the chosen threshold, which balances sensitivity and specificity, and thus, some targets may have escaped our detection. Fig 2A and B provides specific numbers of target genes, and the network structure visualised as a force-directed

layout, chosen subsets of the data are shown in Fig EV2. The periphery of the network is occupied by genes regulated by single TFs, whereas the centre contains coregulated genes (i.e. genes which are downstream of > 1 TF). Large groups of double-regulated targets can be distinguished in between the two zones. We observed large transcriptomic changes for 10 TFs, previously not identified as essential in Hoxb8-FL cells (Basilico *et al*, 2020) (> 200 target genes). Reassuringly, we detected strong effects for several essential TFs, proving that analysis at an appropriate time-point permits the capture of transient cell stages. For more detailed downstream analysis, we focused on the 19 TFs with > 200 targets (essential + non-essential) and considered three aspects of the network: how TFs coregulate their targets, how TFs regulate each other's expression and which target genes form functional modules with common regulatory mechanisms.

## TF coregulation, regulatory modules and common regulatory mechanisms

Focussing on the 19 TFs with > 200 targets, we next calculated gene overlaps and correlations in expression changes for all pairs of TFs to highlight potential functional relationships between them (Fig 3A and B, Appendix Fig S3A and B). As expected, Myc and Max, known to operate in the same complex, share a large fraction of target genes with very high correlation. Of note, not all TFs exhibit strong target overlap as Fos shares only a small fraction of its 228 targets with other factors. Importantly, defining target genes by gene expression changes means that the network model will contain primary and secondary (or higher order) targets. Thus, a target gene with two upstream regulators (TF1, TF2) may receive both inputs in parallel or sequentially. Consequently, if TF1 activates TF2, their shared targets would be expected to change in the same direction. Our network shows examples of such behaviour (see below), but it may not be a universal trait due to other regulatory factors (e.g. a feed forward loop dampening the response) or the time required to manifest secondary and tertiary effects. While resolution of primary and secondary targets is difficult without dynamic data, our network captures some hierarchical regulation, as it contains information on cross-regulation of the 19 TFs (Fig 3C). A case in point is *Cebpa*, a key myeloid regulator and essential for Hoxb8-FL cell survival (Avellino & Delwel, 2017; Basilico *et al*, 2020). We detect 748 genes downstream of Cebpa, including a large number of myeloid factors such as *Irf8*, *Trem3*, *Prtn3*, *Hp* and *Anxa3* (Appendix Fig S2A). A wide range of TFs bind the *Cebpa* locus (Cooper *et al*, 2015; Avellino *et al*, 2016) but their relevance was unclear (Avellino & Delwel, 2017). Our network pinpoints the *Cebpa* regulators *Erg*, *Lmo2* and an unexpected input from *Gata3*. An example of cross-regulation of TFs through core circuits is illustrated by the observation that *Cebpa*, *Gata3* and *Lmo2* coregulate 37 genes, including activation of myeloid genes like *Prtn3*, *Mmp8*, *Ctsg*, *Anxa3*, *Nrg2* and suppression of B-cell genes *Cd79a*, *Mzb1*, *Myl4* or megakaryocytic gene *Cd9* (Fig EV2A, Appendix Fig S2C).

Hoxb8-FL cells rely on Hoxb8 activation to suppress myeloid differentiation (Redecke *et al*, 2013). Interrogation of our network model reveals that *Hoxb8* opposes *Cebpa* as well as other myeloid factors such as *Spi1* and *Myb* (Fig 3A and B). This function of *Hoxb8* appears to be executed at least in part by activating *Hoxa9* and *Meis1*, previously reported anti-myeloid factors (Zeisig *et al*, 2004). We observe strong correlation in target gene expression between *Hoxb8* and *Meis1* and to a lesser extent between Hoxb8 and Hoxa9 (Fig 3A). This involves repression of numerous myeloid factors: Mpo, Prtn3 (regulated by all three factors), Il6ra, Irf8 (Meis1 and Hoxb8), Elane and Hp (Hoxa9 and Hoxb8) (Appendix Fig S2D). Of note, only a limited number of targets were shared between *Meis1* and *Hoxa9* suggesting that they may play complementary roles in suppressing myeloid differentiation. Additionally, the network model highlights a negative correlation between *Tcf3/E2A* and Cebpa and to some extent *Gfi1*. *Tcf3* classically plays a pro-lymphoid role (Boller & Grosschedl, 2014), consistent with *Tcf3* activating lymphoid factors *Gata3* and *Ebf1* in Hoxb8-FL cells. Moreover, *Ebf1* and *Tcf3* coregulated B-cell lineage factors such as *Mzb1* and *Igll1* (Appendix Fig S2F).

*Cbfb*, *Runx1* and *Runx2* are all essential for Hoxb8-FL cell growth, and their targets exhibit high correlation (Appendix Fig S3B), consistent with the known dimerisation of Runx and Cbfb proteins (Warren *et al*, 2000; Yan *et al*, 2004). Of note, Runx/Cbfb targets appear to be involved in promoting myeloid gene expression and antagonise the Hoxb8 programme. For instance, myeloid lineage genes *Mpeg1*, *Afap1*, *Nrp1* and *Dtx4* are activated by Cbfb but repressed by Hoxb8 (Appendix Fig S2B). Interestingly, Runx1/Runx2 and Cbfb show different regulatory patterns with several other factors (Gfi1, Mitf, Rad21; Appendix Fig S3B), suggesting that Runx1/Runx2 and Cbfb may play roles outside of their common protein complex.

In addition to the TF-TF regulation, we identified 47 target gene modules (Dataset EV3). These represent groups of genes with common patterns of regulation by the assayed TFs, for instance modules 6 and 10 are enriched for genes co-activated by Myc, Max and Ebf1, while genes in module 12 are mostly co-activated by Myc/Max/Gfi1 but repressed by Ebf1 (Fig 3D, Appendix Fig S3C–G). For instance, modules 6 and 10 contain genes involved in replication, biosynthesis and mitochondrial biogenesis (enrichment analysis is provided in Dataset EV5) which are co-activated by Myc, Max and Ebf1, highlighting a novel function of Ebf1. On the other hand, module 19, with genes involved in replication and translation (Dataset EV5), is similarly activated by Myc and Max but instead of Ebf1 receive inputs from Cebpa. Importantly, correlation of Myc/Max and Ebf1 targets is not universal and depends on the gene module. Module 12 contains multiple cell cycle genes (*Ccne1*, *Ccnb1*, *Cenpt*, Dataset EV5) activated by Myc and Max but suppressed by Ebf1, suggesting that Ebf1 may play a balancing role between cell growth and proliferation. The module analysis explains to a large degree observed coregulation between Myc/Max and other lineage-specific factors like Cebpa, Ebf1 and Gfi1 presented in

**Figure 2. A functional network connecting 39 transcription factors with their targets.**

A  Number of target genes identified using differential expression for each assayed TF. *genes identified as essential for self-renewal of Hoxb8-FL cells in (Basilico *et al*, 2020).

B  A force-directed graph displaying perturbed transcription factors (orange dots) and target genes (grey dots). Edges indicate if the target gene is differentially expressed, blue for genes downregulated and red for genes upregulated. Size of the nodes is proportional to their degrees.

The EMBO Journal

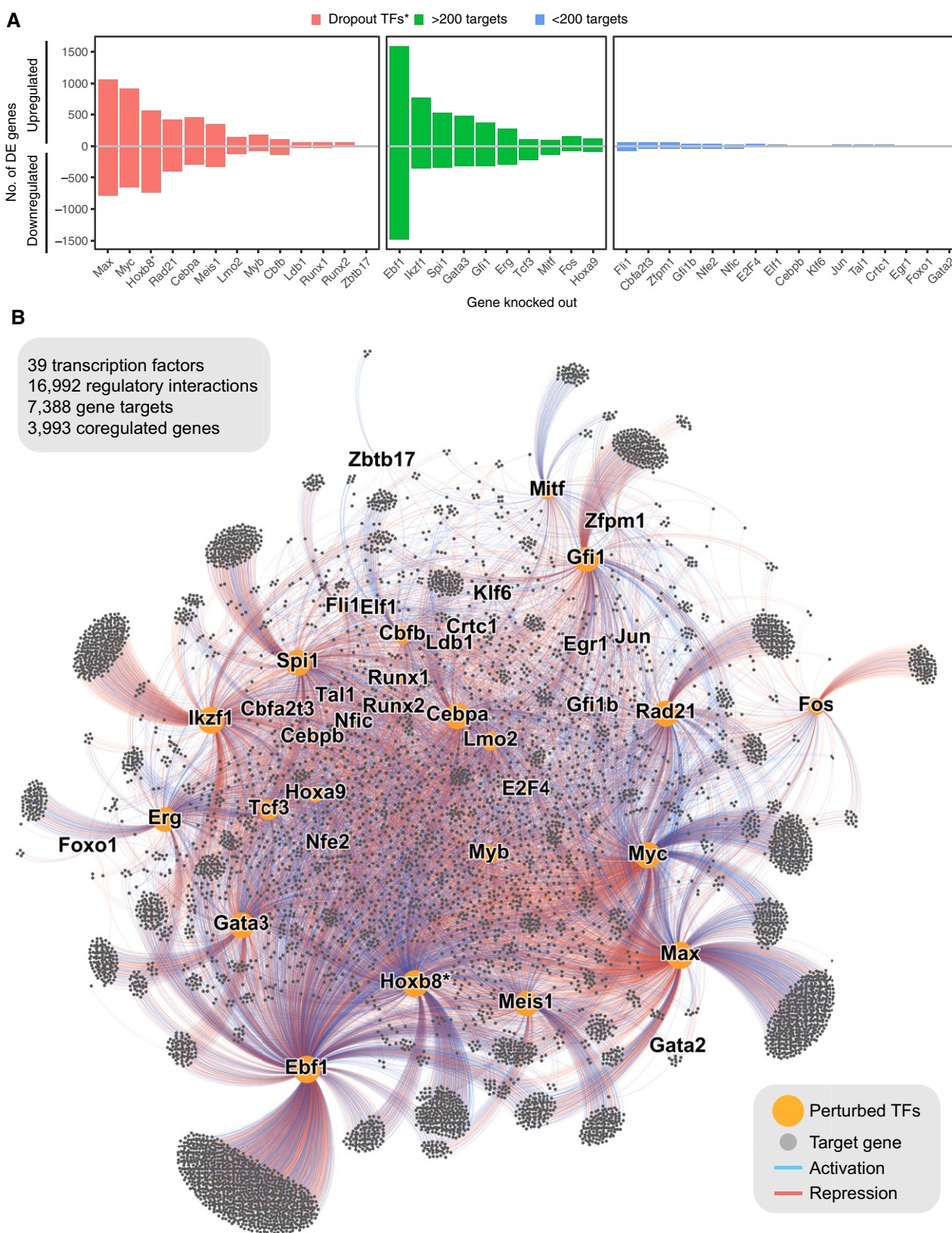

**Figure 2.**

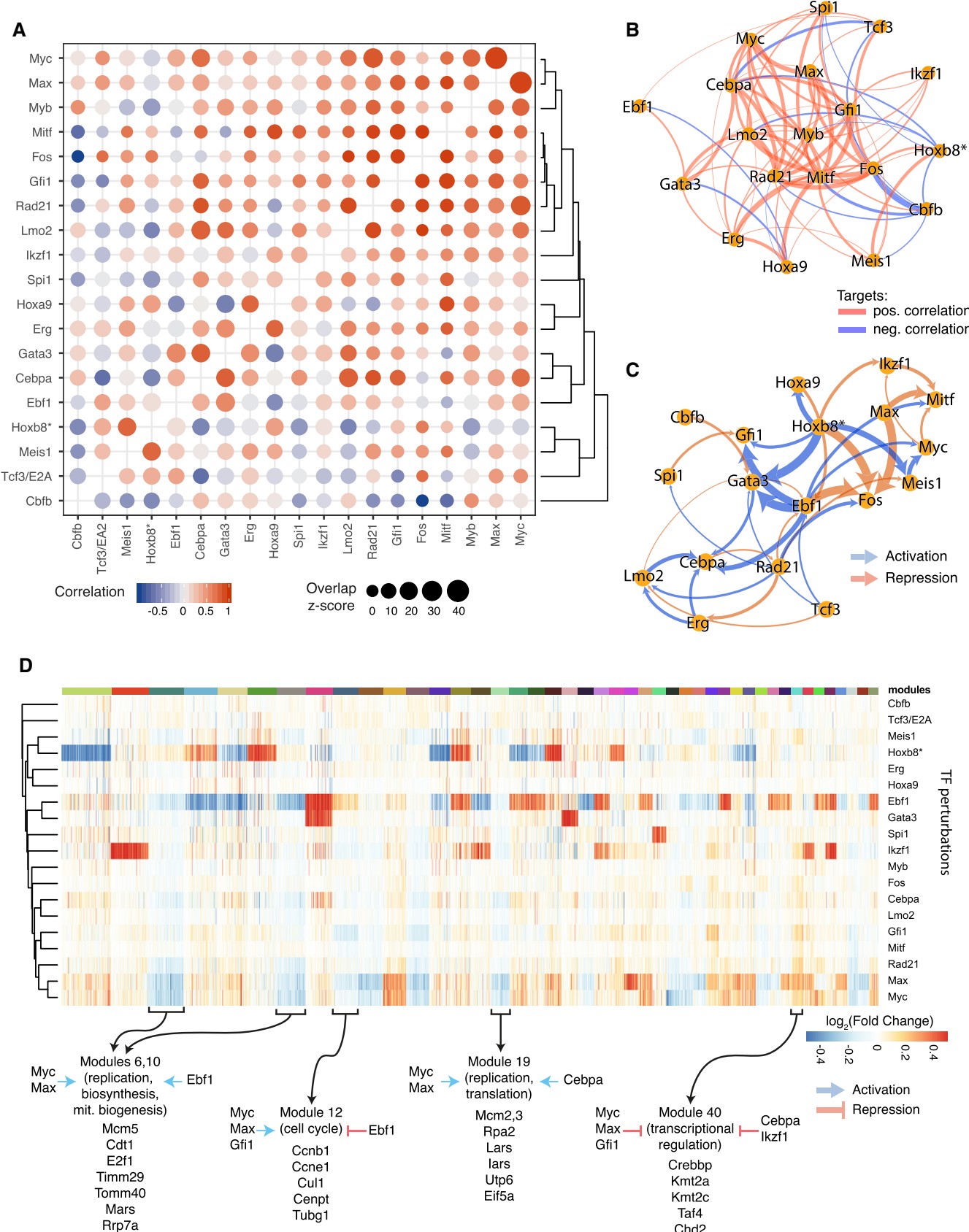

Figure 3.

◀

Figure 3.   **Network analysis provides insight into hierarchy and relations among TFs and their downstream transcriptional programmes.**

A   A representation of the pairwise degree of overlap in targets (size of the circle) and correlation in gene expression changes (colour) among overlapping targets for indicated TF perturbations. Red indicates positive correlations and blue negative correlations.

B   Network view of (A) showing relations among TFs based on their target correlation/anti-correlation. Edge width is proportional to the absolute value of the correlation. To increase readability, connections with |correlation| < 0.4 are not plotted; all correlation values are shown in A.

C   Network view of TF-TF cross-regulation. Directed edges indicate how transcription factors regulate each other's expression. Edge width is proportional to the magnitude of gene expression change (for clarity capped at a value corresponding to absolute $\log_2$(fold change) of 0.8).

D   Identification of target gene modules—groups of genes with common regulatory patterns by TFs. Colour indicates the fold change (adjusted for significance) of gene expression following each TF perturbation. Rows (perturbed TFs) and columns (target genes) are hierarchically clustered. Forty-four modules for target genes are shown. Modules: 1, 2 and 3 are omitted for clarity, all modules are listed in Dataset EV3. Selected modules with highlighted overall regulation pattern and example genes listed below. Gene enrichment analysis for indicated modules is provided in Dataset EV5.

Data information: For clarity, only TFs with > 200 target genes detected are shown in all panels. Data for all TFs are available in Appendix Fig S3A and B. Hoxb8*—gene ectopic expression is disabled by β-oestradiol withdrawal.

Fig 3A and B. Altogether, our network reveals a wealth of relations among transcription factors at a single gene resolution, specific hierarchical TF regulation with novel roles in regulating lineage-specific programmes and target gene modules with common regulatory patterns providing new insight into the combinatorial nature of TF function and downstream biological effects.

### Double perturbations reveal TF interactions

To gain deeper insight into possible interactions between TFs, we performed experiments which simultaneously inactivated two transcription factors. We took advantage of the fact that Hoxb8-FL cells rely on exogenous Hoxb8 (activated by β-oestradiol) to prevent differentiation into dendritic cells. Thus, we chose three TFs with strong target overlap with Hoxb8—Cebpa, Meis1 and Spi1 and performed single and double inactivation experiments (Fig 4A) followed by RNA-Seq.

To analyse these data, we employed a two-factor model with interaction (Fig 4B). The model estimates three coefficients for each target gene, two for expression changes caused by each single perturbation and one as an interaction term (which is the difference between changes caused by the double perturbation and the sum of single perturbations). Thus, a non-zero interaction term indicates a TF-TF relation beyond additive and can point towards a particular mechanism of action. For instance, three positive coefficients indicate synergy (also known as aggravating or synthetic interaction; Segrè *et al*, 2005), meaning that combined perturbation results in a greater effect than the sum of the single perturbations. Conversely, the interaction term with the opposite sign to single perturbations implies a buffering relation (alleviating or suppressive interaction), where the combined perturbation has smaller effect than the sum of its two parts.

Firstly, we focused on the coregulated genes between Cebpa/Meis1/Spi1 and Hoxb8* pairs. We applied low-stringency filtering (|$\log_2$(fold change)| > 0.2) to obtain an overview of the interaction class distributions (Fig 4C–E), and we observed hundreds of genes with potential non-additive regulation. In the previous section (Fig 3C), we predicted that Hoxb8 is an upstream activator of Meis1. Consistently, the Meis1/Hoxb8 interactions mainly belong to the buffering class, which is expected for positively-linked hierarchical factors (Segrè *et al*, 2005; Fig 4D and E). To provide specific gene-level annotation, we applied more stringent criteria (genes DE in each comparison, respectively, i.e. |$\log_2$(fold change)| > 0.2 and FDR < 0.1). We observed a complex pattern of interactions in each case, with non-additive interactions being most common in the case of Cebpa

(Fig EV3C). We provide a detailed overview of genes in each class in Fig EV3A and B, and Dataset EV6 to facilitate further investigation. Overall, combinatorial perturbation data support our previous predictions and reveal common and complex TF-TF interactions in target regulation.

### Genome-wide binding profiles support regulatory network interpretation

The interpretation of regulatory processes that underlie differential gene expression following TF perturbation can be enhanced by the generation of complementary genome-wide TF binding maps. In addition to our previously published maps of open chromatin captured by ATAC-Seq (Basilico *et al*, 2020), we also generated chromatin immunoprecipitation (ChIP-Seq) datasets using Hoxb8-FL cells and antibodies against 14 TFs as well as the H3K27Ac histone modification that indicates transcriptionally active chromatin (Appendix Figs S6 and S7). Thirteen out of 14 analysed TFs exhibit extensive binding across the genome (> 5,000 peaks at *P*-value < $10^6$), and a narrower set of 1,500 sites was observed for Tcf3. The TFs exhibit a remarkably similar distribution across genomic features with most of the binding away from promoters (Appendix Fig S4A), even though the promoter bound fraction was slightly higher for Erg, Runx1, Fli1, Gfi1 and Gfi1b.

Analysis of global binding profiles revealed highly overlapping binding events for Cebpa/Cebpb and Gfi1/Gfi1b, respectively (Fig 5A), in line with their high homology in DNA binding domains (van der Meer *et al*, 2010; Avellino & Delwel, 2017). Furthermore, we observe high similarity of binding events across 5 members of the heptad group (Lmo2, Runx1, Fli1, Erg, Tal1) previously reported to control HSPC genes (Wilson *et al*, 2010). Importantly, there is only a partial agreement in interactions identified by ChIP-Seq and DE. The discrepancy may be in part due to DE capturing also secondary targets, as in the case of Gata3, Lmo2 and Cebpa. Nevertheless, it is not the only explanation as Spi1 and Fli1 co-occupy a large portion of sites, yet share very few targets.

Next, we asked how well neighbouring TF binding sites predict target genes. We compared the observed number of genes simultaneously regulated (DE) and bound (ChIP-Seq) by a given TF, with the number expected from random association (Fig 5B). The observed enrichment was almost uniformly low (below 2), even in the case of TFs with large numbers of targets. Furthermore, peaks corresponding to functionally regulated genes do not appear to have a preferred binding to any genomic features (Appendix Fig S4B). However, genes with nearby Tcf3 peaks strongly associate with those downregulated in DE

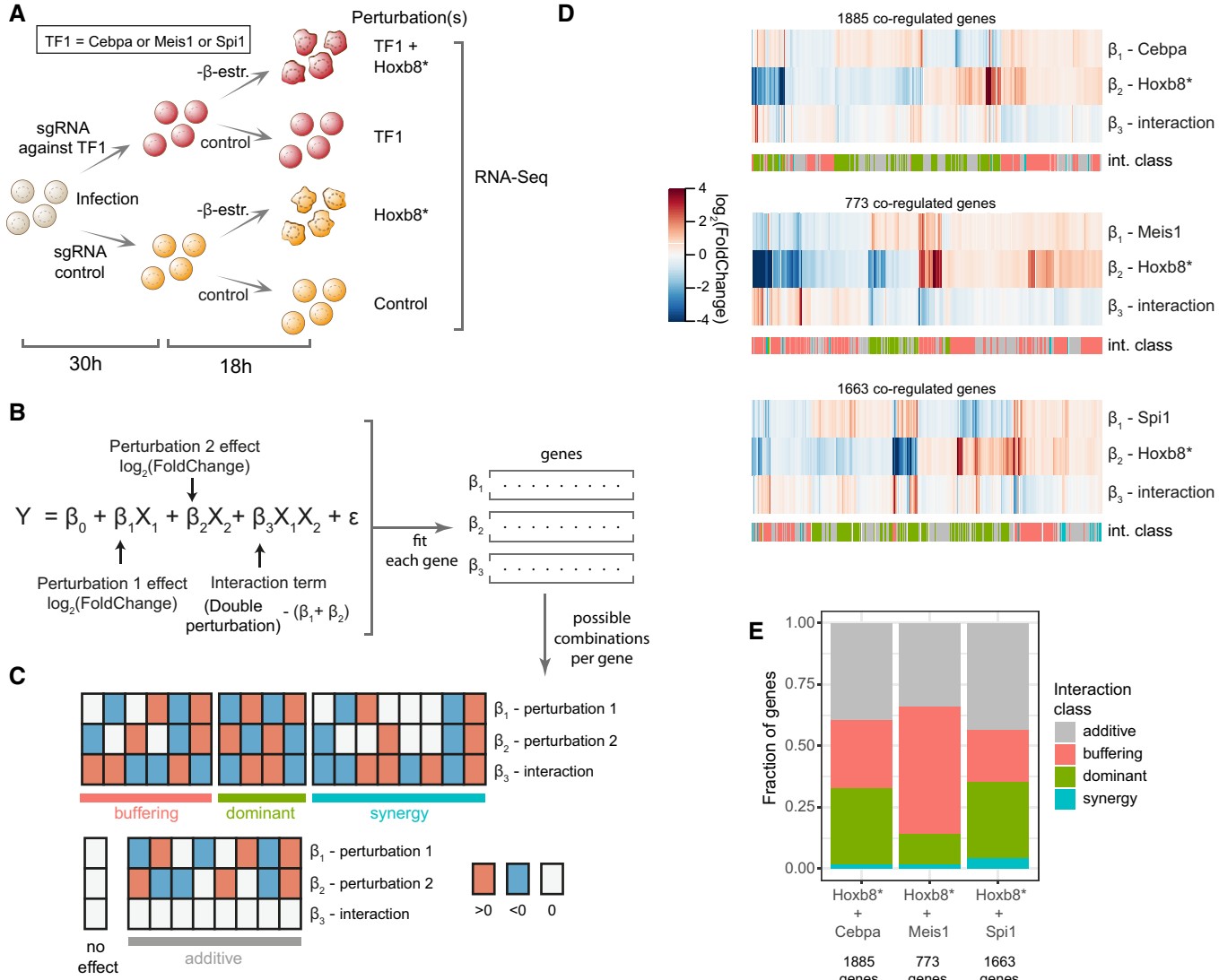

**Figure 4. Double perturbations reveal patterns of interactions between Hoxb8 and other transcription factors.**

A   Experimental design: Hoxb8-FL were transduced with sgRNAs targeting the indicated TF (or a control sgRNA) and subjected to β-oestradiol withdrawal (switching off Hoxb8 ectopic expression—Hoxb8*) or cultured in normal conditions (control).

B   A two-factor linear model with interaction used to fit the data. Observed expression (Y) is modelled as a sum of expression in control cells ($\beta_0$), effect of perturbation 1 (binary factor $X_1$ and coefficient $\beta_1$), effect of perturbation 2 ($X_2$ and $\beta_2$), their interaction ($X_1X_2$ and $\beta_3$) and the error term ($\sigma$). The interaction term can be interpreted as the difference between the expression changes in the double-perturbed cells and the sum of coefficients of $\beta_1$ and $\beta_2$.

C   Binary combinations of directions in observed expression changes for perturbation 1, perturbation 2 and the interaction term grouped into four general classes. Based on classification implemented in Dixit *et al* (2016).

D   Changes in expression for genes coregulated by separate Cebpa/Meis1/Spi1 and Hoxb8 perturbations (FDR < 0.1 and |log$_2$(fold change)| > 0.2). The interaction row indicates changes beyond simple additive effect (white = additive effect). Each gene was annotated with an interaction class (int. class) as explained in (C), using the |log$_2$(fold change)| > 0.2 threshold to assign change direction.

E   Fractions of genes in each interaction class showed in (D).

analysis, regardless of the relative positions of peaks in gene elements. Thus, Tcf3 appears to act as a strong activator of expression, with binding alone being a strong indication of functional regulation. To enable further analysis of specific regions, e.g. highlighting TF-target primary interactions, we provide an interactive UCSC session (http://genome-euro.ucsc.edu/s/idk25/TFnet2020_allChIPs_impr).

Finally, we analysed binding profiles for the regulators of *Cebpa* and Gata3 identified in the previous section. The inferred novel

regulation of *Cebpa* expression by Gata3, Lmo2 and Erg is supported by clear binding of these factors downstream of the locus, specifically nearby a previously identified + 37 kb enhancer, critical for *Cebpa* expression and myeloid differentiation in mice and humans (Fig 5C; Cooper *et al*, 2015; Avellino *et al*, 2016). The *Gata3* downstream regulatory region contains a known enhancer (Tce1) (Hosoya-Ohmura *et al*, 2011; Ohmura *et al*, 2016) critical for Gata3 function in T-cell maturation. In our data, we observed extensive

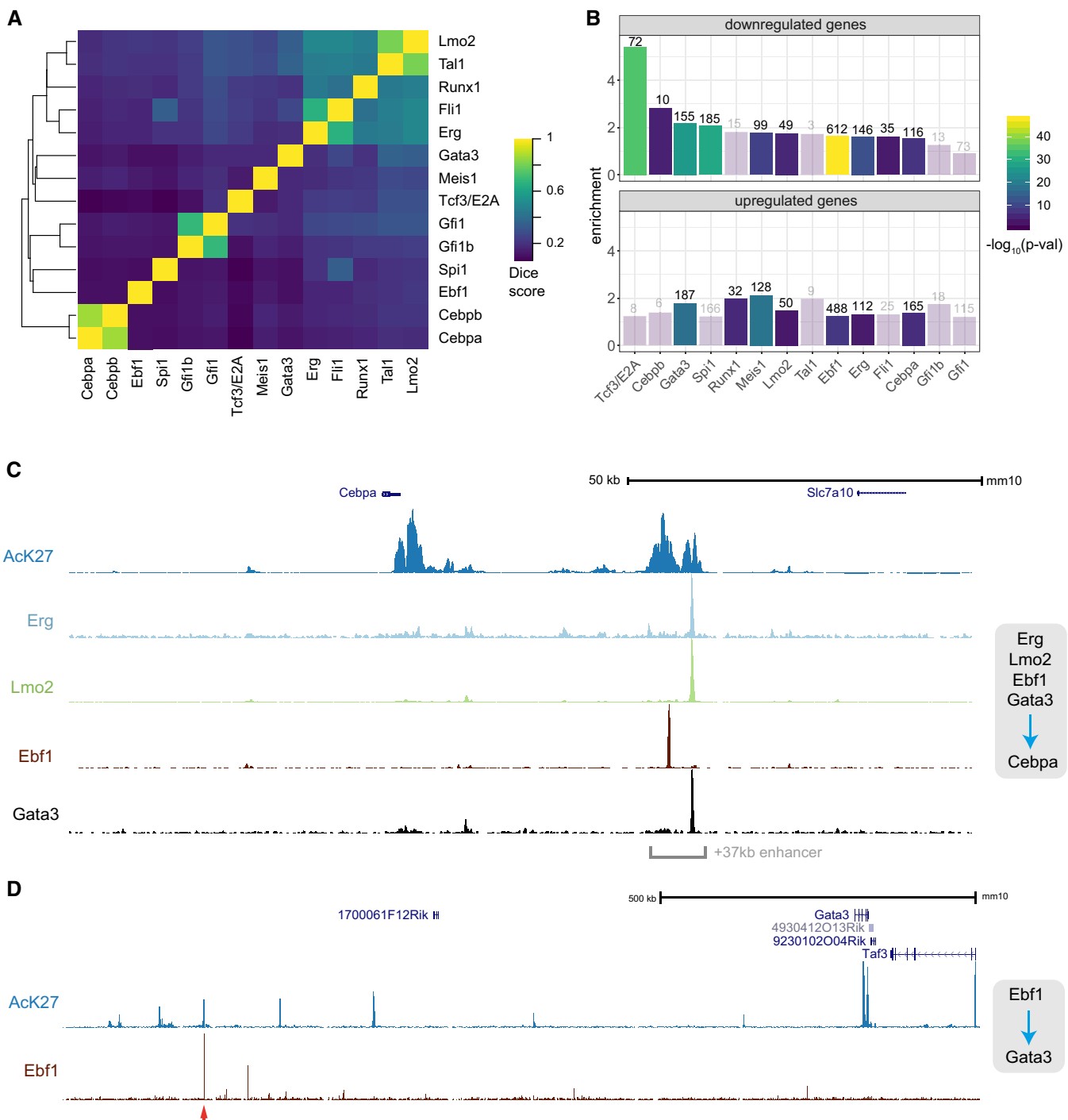

**Figure 5. Genome-wide annotation of chromatin states highlights putative primary transcription factor targets.**

A    Dice scores summarising overlaps among peaks identified for 14 ChIP experiments in Hoxb8-FL cells.

B    Comparison of genes identified as differentially expressed following TF loss with mapped ChIP peaks in a corresponding experiment. Enrichment over a random value and the *P*-value (based on the hypergeometric test) are shown (bars corresponding to non-significant enrichment values are semi-transparent). Analysis was performed separately for genes up- and downregulated.

C, D    Genomic views of relevant TF binding to Cebpa and Gata3 loci (supporting direct regulatory interactions identified in Figs 3A–C). The red arrow indicates a putative enhancer element, which is bound by Ebf1 and flanked by AcK27-rich regions.

epigenetic marks and putative regulatory elements stretching hundreds of kilobases downstream of Gata3 (Fig 5D). These included binding by Ebf1 and multiple other factors (like Cebpa and Spi1), as well as H3K27ac-rich regions. This is in agreement with Gata3 receiving multiple regulatory inputs (Fig 3C) and its expression strongly activated by Hoxb8 and Ebf1. We explore this previously unknown relationship between Gata3 and Ebf1 below.

### Hoxb8-FL cells employ Ebf1 to balance growth and self-renewal

Gata3 and Ebf1 are usually seen as antagonistic factors, with their respective upregulation associated with establishing either a T- or B-cell fate. This is reflected by reciprocal expression patterns in B- and T-cell lineages (Fig 6A and B). Of note, *Ebf1* expression is barely detectable in the most immature cells, while *Gata3* shows high expression among the HSCs with a gradual decrease towards more committed cells with the exception of T cells (Fig 6A and B). Indeed, *Gata3* has previously been reported to control HSC self-renewal and cell cycle, although there is as yet no consensus on the exact role of *Gata3* in HSCs (Buza-Vidas *et al*, 2011; Ku *et al*, 2012; Frelin *et al*, 2013). *Ebf1* function has so far been mostly studied in the context of lymphoid development (Boller & Grosschedl, 2014; Boller *et al*, 2018), where it is critical for promoting B-cell differentiation and commitment while suppressing other cell fates.

Hoxb8-FL cells simultaneously express *Gata3* and *Ebf1* (Fig 6A), we unexpectedly find that *Ebf1* activates *Gata3* expression (Fig 3C), and their downstream targets tend to correlate (Fig 3A). To investigate the cause of such correlation, we investigated genome-wide chromatin data. The ChIP-Seq binding analysis (Fig 5A) indicates that Gata3 and Ebf1 do not preferentially bind common regions. To further support this, we analysed our previously published open chromatin (ATAC-Seq) data (Basilico *et al*, 2020) and identified ~300 footprints for Gata3 and Ebf1 each (Appendix Fig S5A and B). Among Ebf1 footprints and RNA-Seq targets, we identified 86 common genes, that are preferentially downregulated following Ebf1 loss (Appendix Fig S5D, Dataset EV7); in case of Gata3, the number of such genes was limited (Appendix Fig S5D). Consistently, with the ChIP-Seq data, Gata3 and Ebf1 footprints were almost entirely exclusive (Appendix Fig S5C). We extended our ChIP-Seq annotation with these footprinting data to provide another layer of information for further studies. In conclusion, Ebf1 and Gata3 do not seem to bind sequences in a coordinated manner, and the Ebf1-Gata3 activation may largely explain the apparent correlation among their shared targets. Nevertheless, among Gata3 targets with a nearby Gata3 binding site, 108 out of 475 are also associated with an Ebf1 site.

Thus, we cannot exclude that in some cases Gata3 and Ebf1 may directly coregulate a target gene from separate binding sites.

To dissect Gata3 and Ebf1 interplay further, we investigated both concordant and discordant gene expression changes of their shared downstream targets (Figs 6C, Appendix Fig S2G–J, Dataset EV4, annotation provided in Fig 7B and C). Sixty-seven genes are repressed by Gata3 and activated by Ebf1, containing numerous genes associated with the B-cell programme (*Cd79a, Cd79b, Vpreb1, Vpreb3*; Appendix Fig S2J), suggesting that the role of Ebf1 in asserting a B-cell programme is indeed kept in check by Gata3 (Banerjee *et al*, 2013; García-Ojeda *et al*, 2013; Nechanitzky *et al*, 2013). More surprisingly, multiple genes are activated by both Gata3 and Ebf1, several of which are associated with the myeloid programme (*Prtn3, Mpo, Ctsg*; Appendix Fig S2I), a function executed at least in part by Cebpa, predicted by our network to be downstream of both Gata3 and Ebf1 (Fig 3C). Furthermore, Ebf1 appears to serve a dual role in regulating lymphoid genes. Contrary to the activation of genes associated with a later B-cell programme outlined above, Ebf1, and also Gata3, represses genes involved in early lymphoid steps (common to B and T cells) such as *Il7r, Flt3, Tcf4* or *Rag1* (Appendix Fig S2G), suggesting that Gata3 and Ebf1 act together to maintain self-renewal and prevent premature expression of the lymphoid programme, while promoting the myeloid one.

From our analysis, Ebf1 emerges as a hub controlling expression of not only lineage programmes but also gene modules involved in DNA replication, biosynthesis and cell cycle. As we did not originally identify *Ebf1* as an essential gene (Basilico *et al*, 2020), we followed closely Hoxb8-FL cells after inactivating *Ebf1* (Fig EV4A and B). Initially, cells increased their proliferation rates, which was accompanied by a reduction in cell size and later followed by slower growth and outcompetition from culture by WT cells. As a control, we inactivated Myc in Hoxb8-FL cells, which quickly caused slower growth and smaller cell size as expected. This initial over-growth of Ebf1-deficient cells is most likely responsible for our failure to detect Ebf1 as essential in the original CRISPR/Cas9 screen. The enhanced proliferation accompanied by smaller cell size in the absence of Ebf1 suggests that *Ebf1* limits cell cycle rate, and without it, cells may not be able to accommodate increased growth demand. This is in line with Ebf1 inhibition of cell cycle genes (module 12, Fig 3D) inferred from the gene expression data alone.

To identify similar Ebf1-dependent programmes in other cell types, we cross-compared Ebf1 binding profiles across available datasets. The Ebf1 signature in Hoxb8-FL cells is clearly distinct from late B-cell differentiation stages but shows high similarity to a transient cell state following *Ebf1* re-expression in Ebf1$^{-/-}$ pre-pro-B

---

**Figure 6. TF network uncovers novel functions of Ebf1 and Gata3 in maintaining a multipotent, self-renewing state.**

A, B   Bulk RNA-Seq expression levels for Gata3 and Ebf1 in a set of isolated primary cells (data from Immgen (Heng & Painter, 2008)) compared to Hoxb8-FL cells. Horizontal line indicates mean expression.

C   Correlation in gene expression changes following Ebf1 and Gata3 loss in Hoxb8-FL cells. Each quadrant corresponds to a set of target genes with a common regulation by Gata3/Ebf1-I—inhibition/inhibition, II—activation/inhibition, III—activation/activation, IV—inhibition/activation. For each group, the sums of scaled expression values in mouse LK/LSK or human BMMC landscapes is plotted to highlight cell types with the highest overall expression. For landscape annotation, see Fig 7B and C. Changes in expression for example genes are provided in Appendix Fig S2G–J. Blue line indicates the linear fit with shaded areas as confidence intervals.

D   Schematic representation of the experiment performed by (Li *et al*, 2018). Ebf1 was re-expressed in Ebf1$^{-/-}$ pre-pro-B cells thus resuming their differentiation.

E   Correlation in expression changes after 24 h following *Ebf1* loss in Hoxb8-FL cells and re-expression of Ebf1 in *Ebf1*$^{-/-}$ pre-pro-B cells (Li *et al*, 2018). Blue line indicates the linear fit with shaded areas as confidence intervals.

F, G   Expression of the key marker genes (F) and *Ebf1, Pax5, Gata3* and *Cebpa* factors (G) along pseudotime corresponding to the B-cell differentiation trajectory in human foetal liver cells, data from Popescu *et al* (2019).

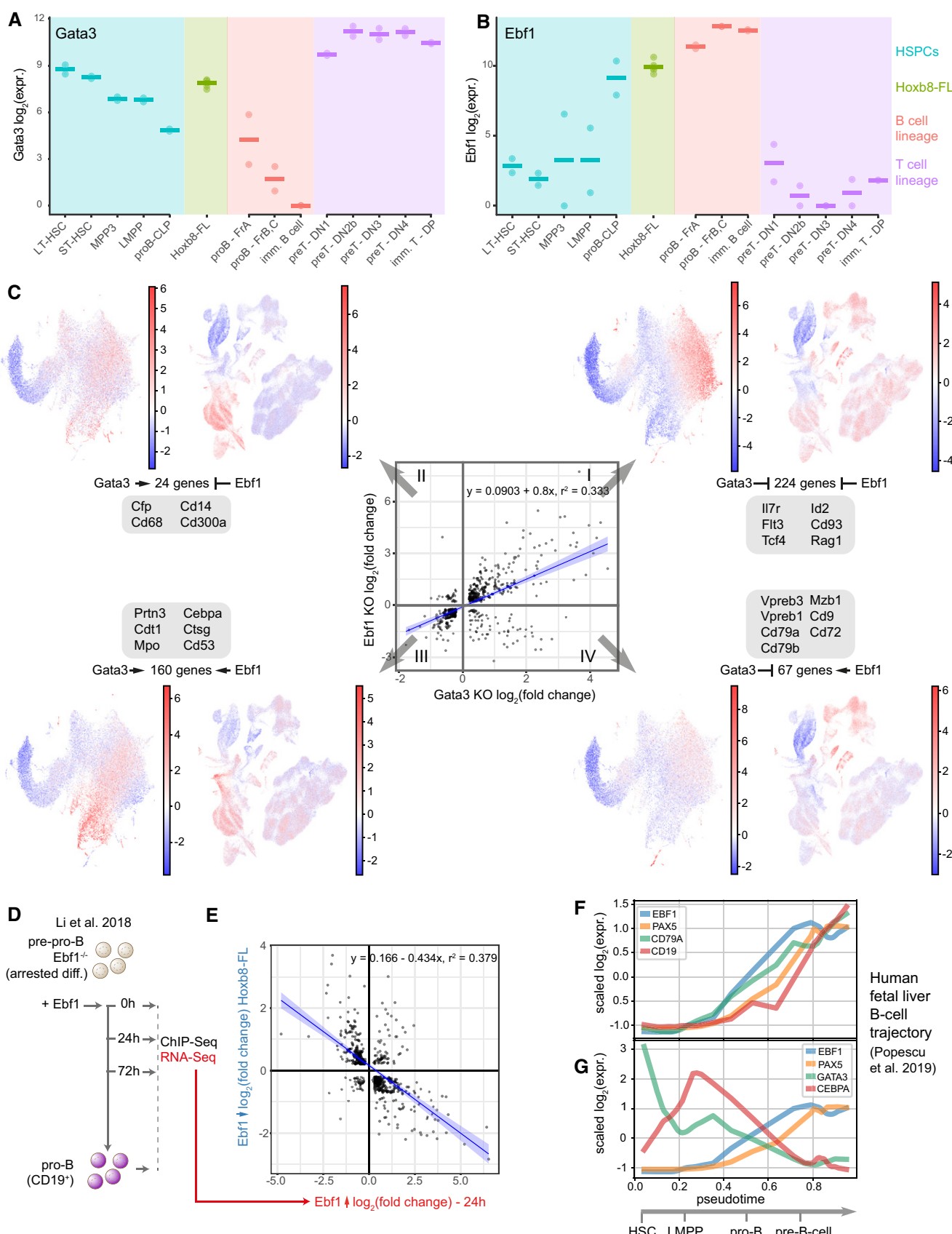

Figure 6.

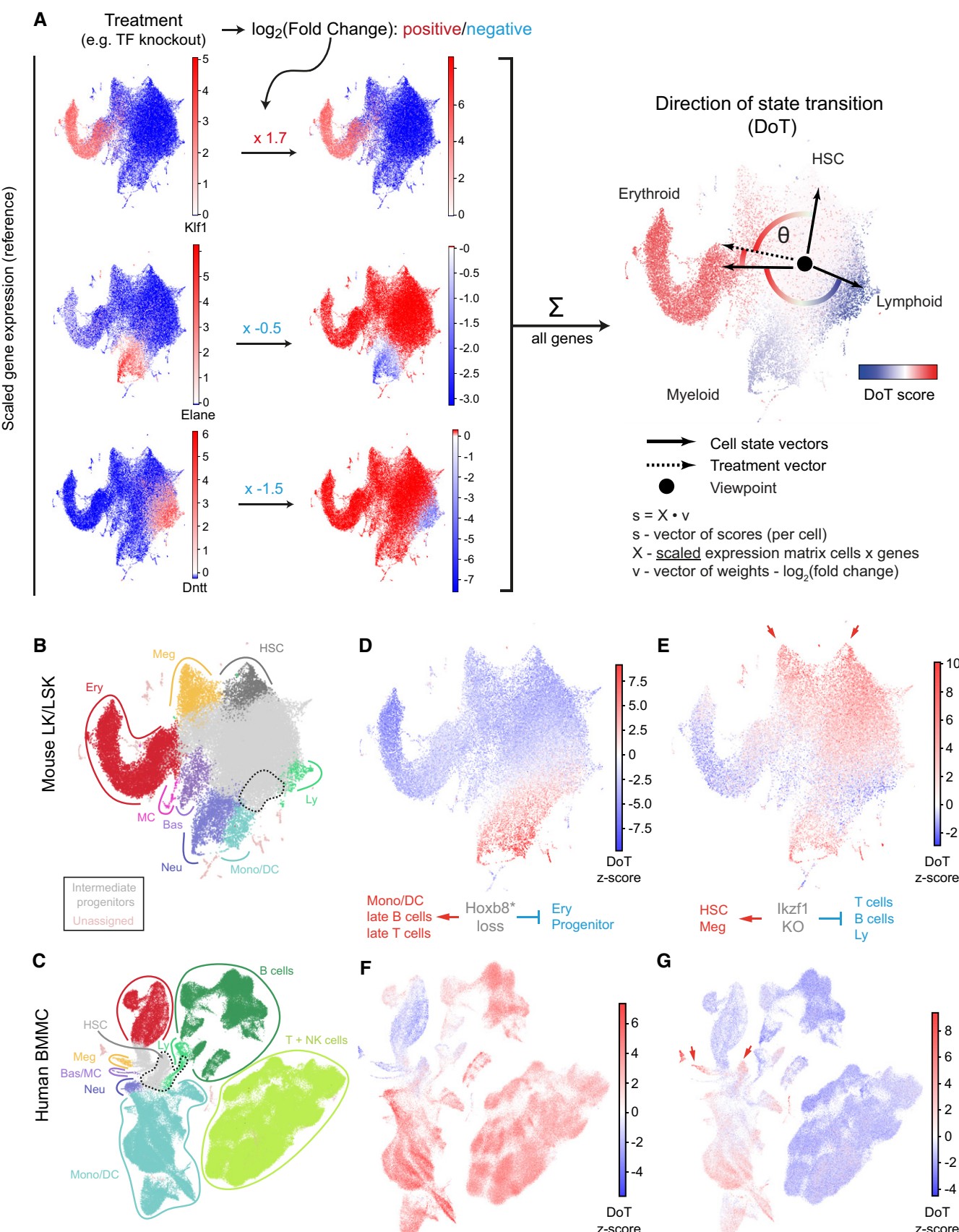

**Figure 7.**

**Figure 7. Understanding transcriptomics changes following a TF perturbation with scRNA-Seq landscapes.**

A    A toy example with visual explanation of the direction of state transition (DoT) calculation. Three lineage-specific genes are used (full-scale DoT analysis is unbiased and uses all available genes): erythroid Klf1, neutrophilic Elane and lymphoid Dntt; their scaled expression is plotted on the reference landscape (annotation in (B, C)). Upon treatment, changes in expression are observed for each gene. We calculate contributions from each gene as the product of its scaled expression and changes in expression (log$_2$(fold change)). DoT score is the sum of these components and indicates direction of cell state transition with respect to the chosen viewpoint (point of origin).

B, C    Annotated UMAP projections of scRNA-Seq landscapes—mouse LK + LSK populations (Dahlin *et al*, 2018) and Human Cell Atlas bone marrow mononuclear cells (Regev *et al*, 2017). Dashed lines indicate cluster positions, which mean expression values were used to designate points of origin for the DoT score analysis.

D, E    DoT scores calculated using genes differentially expressed after Hoxb8 or Ikzf1 loss in Hoxb8-FL cells in the context of mouse LK/LSK landscape (Dahlin *et al*, 2018).

F, G    DoT scores calculated using genes differentially expressed after Hoxb8 or Ikzf1 loss in Hoxb8-FL cells in the context of the human BMMC landscape. Arrows indicate the HSCs and megakaryocytic trajectories.

cells (i.e. resuming differentiation) (Li *et al*, 2018; Figs 6D and EV4C). What is more, the transient gene expression changes caused by Ebf1 re-expression overlap and negatively correlate with the ones observed after *Ebf1* knockout in Hoxb8-FL cells (Figs 6E and EV4D). Importantly, this overlap includes Ebf1 downstream targets associated with a myeloid expression programme (Fig EV4E and F). Of note, the differentiating pre-pro-B cells transiently co-express the key lineage factors *Ebf1*, *Cebpa*, *Gata3* and *Pax5*, with *Cebpa* and *Gata3* suppressed as cells progress towards the pro-B stage (Fig EV4G). A self-renewing and multipotent cell population of Pax5$^{-/-}$ pro-B-cells (Nutt *et al*, 1999; Revilla-i-Domingo *et al*, 2012) also co-expresses *Cebpa*, *Gata3* and *Ebf1*, at similar levels to the ALP fraction of the bone marrow (Fig EV4G). Hoxb8-FL cells mirror this state by maintaining high expression of *Cebpa*, *Gata3* and *Ebf1*, but keeping *Pax5* efficiently suppressed by action of *Gata3*.

As the Hoxb8-FL cells share multiple features with progenitors in the early stages of lymphoid differentiation, we searched for a respective progenitor cell state co-expressing *Ebf1*, *Gata3* and *Cebpa in vivo*. scRNA-Seq from human foetal liver (Popescu *et al*, 2019) faithfully reconstructs the early lymphoid/B-cell differentiation trajectory with stereotypical progression of key marker genes (Fig 6F). Pseudotime ordering of the single cells along this differentiation trajectory shows a decrease in *Gata3* expression as cells leave the HSC compartment, but *Gata3* expression remains detectable and is not immediately extinguished as Ebf1 starts being expressed (Fig 6G). Of note, the developmental window where both factors are expressed is additionally accompanied by increasing Cebpa expression. This state, however, is quickly resolved as *Pax5* becomes expressed, with continuous increase of *Ebf1* expression while *Gata3* and *Cebpa* become almost completely undetectable (Fig 6G). We observe matching expression patterns on a smaller sample of lymphoid progenitors from mouse bone marrow (Fig EV4H and I; Loughran *et al*, 2017). Therefore, Hoxb8-FL cells seem to represent an intermediate lympho-myeloid progenitor stabilised *in vitro*, which already expresses potent lineage determinants (Ebf1) but still retains some expression of other lineage factors (*Gata3*, *Cebpa*) and self-renewal genes (*cKit*). Comparison with primary cells thus validates the use of Hoxb8-FL cells for investigating regulatory interactions between cell fate determining TFs.

## Transcriptomic landscape interpretation of gene function

A common endpoint for genome-scale studies is over-representation analysis of lists of up- and downregulated genes, using external databases of gene annotation (e.g. gene ontology). While undeniably useful, ontology categories are arbitrary, as they are dictated by

our language and do not indicate whether effects are positive or negative. Moreover, gene ontology annotations are incomplete and contain errors. To circumvent these issues, we developed a method using scRNA-Seq landscapes as references and show how such an approach transforms our ability to interpret large-scale perturbation studies using our 39 TF knockout screen as an example.

Taking an individual cell within a single-cell landscape as a viewpoint, scaling gene expression in all of the other cells relative to the viewpoint cell creates a set of directions or vectors connecting all other cells to the viewpoint cell. By extension, if we consider a perturbation being applied to the viewpoint state, the observed changes in expression are a direction (perturbation vector) between the original and the new cell state. The degree of alignment (angle) between the perturbation vector and the cell state vectors can then be seen to indicate the direction on the landscape in which cells would shift due to the applied perturbation. To estimate these values, we developed the direction of transition method (DoT) method, building on the Fast-Project gene signature methodology (Fig 7A) (DeTomaso & Yosef, 2016). Our method assigns a DoT score to each single cell which is visualised on a reference scRNA-Seq landscape. High positive values (red) indicate that the perturbation would push the origin cell towards those red single cells, whereas negative values (blue) indicate transition away from those cells. DoT score significance is estimated by calculating z-scores based on simulations. In addition to the score for each cell, the method provides ranked contributing genes, highlighting the relevant downstream targets. As the DoT vectors are arrows in gene expression space anchored at the viewpoint, the DoT score does not follow the complex cell state manifold. However, it can easily be interpreted locally in chosen regions of differentiation trajectories inferred by other methods e.g. PAGA (Saelens *et al*, 2019; Wolf *et al*, 2019).

To explore the differentially expressed gene lists from our KO screen, we chose cells within the mouse and human HSPC scRNA-Seq landscapes most resembling Hoxb8-FL cells as viewpoints, as highlighted together with cell type annotation in Fig 7B and C. Since loss of Hoxb8 activation is known to promote dendritic cell differentiation of Hoxb8-FL cells (Redecke *et al*, 2013), we tested the DoT tool first with the genes up/downregulated shortly after β-oestradiol withdrawal (which equals loss of Hoxb8 activation). Cells located in the monocyte/DC corner of the landscape showed strongly positive DoT scores (Fig 7B, C, D and F), indicating a shift of perturbed cells towards that transcriptional state and thus validating the approach. In addition to a few selected examples discussed below, analogous visualisation of the transcriptional consequences following KO of all other tested TFs is provided in Appendix Figs S8–S13.

As predicted in a previous section based on known functions of selected genes, expression changes downstream of Hoxb8, *Meis1*

and *Hoxa9* KO indicate a shift towards myeloid differentiation (Fig EV5D and E). Interestingly, individual TFs highlighted monocytic/DC and neutrophilic trajectories, respectively, suggesting complementary functions in blocking the two major myeloid differentiation trajectories. Conversely, a strong shift away from the myeloid programme is observed when perturbing known myeloid regulators such as *Cebpa*, *Gfi1* and *Spi1* (Fig EV5A and B, Appendix Figs S9 and S12), with *Gfi1* being more specific towards the neutrophilic trajectory and *Spi1* towards the monocytic/DC lineages. Inactivation of *Gata3*, *Ebf1*, *Lmo2* and *Erg* regulators of *Cebpa* expression identified in this work, also causes a similar shift (Fig EV5C and F, Appendix Figs S8, S9, S11 and S12). This analysis therefore directly reinforces the myeloid subnetwork identified through bioinformatic analysis of our KO screen results.

KO of *Myb* and *Ikzf1* caused gene expression changes consistent with a move from the Hoxb8-FL state to less mature cells, indicating a shift towards the trajectory connecting HSCs and megakaryocyte progenitors (Fig 7E and G, Appendix Figs S9 and S12). There is a limited overlap between the lists of contributing genes suggesting that the two factors have complementary but independent functions. The *Ikzf1* score is particularly high, and indeed, among genes upregulated following *Ikzf1* KO are key HSC and megakaryocyte marker genes *Procr* and *Pf4* (Wilson *et al*, 2015; Dahlin *et al*, 2018). Given that the *Ikzf1* and *Myb* related shifts to HSCs and megakaryocyte progenitors are based on distinct sets of genes, the two genes can have distinct effects elsewhere in the landscape, as illustrated by a dramatic shift away from the B- and T-cell states only seen with *Ikzf1*.

We also used DoT to investigate the regulation of the less commonly studied basophil and mast cell differentiation trajectories, which are predicted to be suppressed by myeloid factors like Cebpa, Spi1 but also Myc/Max, Rad21 and Myb (Appendix Figs S8–S13). Moreover, the basophil programme appears to be activated by Gata3, indicating additional, Cebpa-independent, fate control function of Gata3 (Fig EV5C). Therefore, basophil and mast cell programmes exhibit distinct regulatory patterns from the neutrophilic/monocytic/DC fates, in line with recent reports (Dahlin *et al*, 2018; Tusi *et al*, 2018; Weinreb *et al*, 2020) suggesting an earlier than anticipated separation of the two lineages. Taken together, the DoT score method provides a streamlined interpretation of gene expression changes generated by a variety of techniques, that will be broadly applicable to single-cell landscapes across organisms and tissues. When applied to data generated on shorter time scales (single hours), our method should aid interpretation of nascent RNA data (e.g. scSLAM-Seq or scEU-Seq) (Erhard *et al*, 2019; Battich *et al*, 2020) and complement other cell state prediction techniques such as RNA velocity (La Manno *et al*, 2018). As exemplified above, we utilised this approach to infer (i) direction of cell state shifts, (ii) new biological functions for perturbed factors and (iii) highlight downstream targets relevant for specific biological processes, thus providing comprehensive biological interpretation of the newly generated haematopoietic TF network.

## Discussion

Deciphering gene regulatory networks remains a major challenge, due to the limitations of inferring relations from correlative evidence

and lack of systematic functional data. Here, we show how CRISPR/Cas9 perturbation combined with RNA-Seq readout can be used to construct a functionally defined TF network for haematopoietic progenitors. This network (i) provides nearly 17,000 connections between 39 TFs and their targets, (ii) establishes TF coregulation at common target genes, (iii) unravels regulatory hierarchies among TFs and (iv) organises target genes into modules with common regulatory patterns, highlighting relevant biological functions. Moreover, identification of a surprising role for *Ebf1* and *Gata3* in contributing to a myeloid expression programme illustrates the utility of the network for discovering new biological mechanisms, suggesting that it will constitute a significant resource for future analysis and modelling, as well as serving as a much-needed reference for cross-validation.

The analysis presented here is consistent with the notion that Hoxb8-FL cells reflect a rare and transient state during early myeloid–lymphoid differentiation, where key lineage TFs are co-expressed, reflecting their potential to produce both lymphoid cells and myeloid cells. Importantly, our network analysis now provides insights into the molecular processes that underpin this poised multipotent state. We demonstrate that Hoxb8-FL cells rely on *Ebf1* to control cell cycle rate, which at the same time activates an early B-cell differentiation programme, consistent with previous findings (Györy *et al*, 2012; Boller & Grosschedl, 2014). Differentiation towards the B-cell fate is kept in check by the activity of Gata3, efficiently suppressing the expression of key B-cell factor, *Pax5*. This creates a state similar to that of $Pax5^{-/-}$ pro-B cells, a self-renewing population with both lymphoid and myeloid potential (Nutt *et al*, 1999; Heavey *et al*, 2003; Revilla-i-Domingo *et al*, 2012) which also co-expresses *Ebf1*, *Cebpa* and *Gata3* (albeit the latter at lower levels than in Hoxb8-FL cells) (Fig EV4G). Our experiments show that *Gata3* expression is activated through *Tcf3/E2A* and exogenously expressed Hoxb8 (Fig 3C). As several *Hoxb* and *Hoxa* genes exhibit high expression in HSCs (Pineault *et al*, 2002; Argiropoulos & Humphries, 2007; Nestorowa *et al*, 2016), we speculate that these may also be responsible for high *Gata3* expression in the upper tiers of the haematopoietic hierarchy. Interestingly, *Ebf1* is not counteracting this effect in Hoxb8-FL cells but instead activates *Gata3* expression (this may be direct or indirect), thus reinforcing this primed cellular state. A non-antagonistic relation between *Ebf1* and *Gata3* is consistent with the early steps of a lymphoid/B-cell trajectory that we inferred from *in vivo* gene expression data (Loughran *et al*, 2017; Popescu *et al*, 2019). Early during lymphoid differentiation, *Ebf1* upregulation does not immediately cause *Gata3* downregulation (Fig 6F and G, and EV4H and I). Complete *Gata3* suppression takes place only at higher *Ebf1* levels, coinciding with *Pax5* activation. High Ebf1 levels, e.g. upon overexpression (Banerjee *et al*, 2013), have been shown to suppress *Gata3* expression, but as evident from data reported by Li *et al* (2018) this process occurs slowly over several days (Fig EV4G). Therefore, the introduction of Hoxb8 into primary bone marrow cells is able to establish a TF network wired to pause B-cell differentiation at an early stage while providing a strong growth cue via Ebf1. Although this introduces high Gata3 levels compatible with promoting T-cell differentiation, this route remains unavailable until external Notch is supplied, as would be the case upon entering the thymus.

Hoxb8-FL cells employ complex gene regulation, intertwined with the machinery described above, to prevent myeloid differentiation.

As indicated in Fig 6C and confirmed by the DoT score analysis (Fig EV5C and F), Gata3 and Ebf1 activate expression of a myeloid gene programme, while suppressing the earliest lymphoid markers. Consistently, Gata3 overexpression can promote myeloid differentiation of $Pax5^{-/-}$ cells (Heavey *et al*, 2003). As evident from our analysis of the Hoxb8-FL gene expression state, lympho-myeloid progenitors may activate low levels of both myeloid and lymphoid programmes based on the co-expression of lineage-affiliated factors. Furthermore, given the inferred TF hierarchy (Fig 3C), we expect that the Ebf1 pro-myeloid effect is largely mediated through Gata3 and subsequently Cebpa, a key factor promoting myeloid differentiation (Avellino & Delwel, 2017). Indeed, Cebpa upregulation is a common feature in our analysis of the early lympho-myeloid trajectory as cells leave the HSC territory (Fig 6G). The network presented here also shows that Cebpa expression promotes the cell growth programme (Fig 3D), and its loss decreases the survival of Hoxb8-FL cells Fig EV1G. Interestingly, Cebpa has been previously implicated in inhibiting proliferation of myeloid progenitors (Porse *et al*, 2005), suggesting that Cebpa may play different roles at various stages of differentiation or depending on the cell growth conditions. Finally, Cebpa receives positive inputs from other self-renewal factors such as Erg and Lmo2 thus highlighting its role as crucial hub.

Importantly, Hoxb8-FL cells must prevent myeloid differentiation in order to maintain a self-renewing culture. Our analysis highlighted Meis1 and Hoxa9 as two factors with established roles of blocking myeloid differentiation, most notably in the context of acute myeloid leukaemia cells (Zeisig *et al*, 2004). Accordingly, DoT score analysis confirms that Hoxb8-FL cells activate the myeloid programme following inactivation of either *Hoxa9* or *Meis1*. Interestingly, Meis1 appears to be more specific towards monocytic/dendritic cell lineages, while Hoxa9 acts more on the neutrophil programme. This suggests that despite being reported as parts of the same complex (Shen *et al*, 1999), the two proteins can serve at least partially complementary functions in preventing myeloid differentiation. While Hoxa9 is not essential, Meis1 and Hoxb8 maintain self-renewal of Hoxb8-FL cells and indeed both prevent upregulation of *Irf8*, a factor responsible for monocyte and dendritic cell differentiation (Yáñez & Goodridge, 2016). Altogether, our data reveal a fascinating interplay between multiple co-expressed lineage factors. These, in agreement with their established functions, drive-specific cell expression programmes but we show here how their specific wiring at single gene resolution ensures that no lineage becomes dominant and cells maintain their multipotent state, while providing sufficient growth signals.

The Waddington landscape is a powerful analogy, but its real-world application requires detailed understanding of the landscape shape (to connect cell states with their fates) and the complex regulatory mechanisms underneath (to modulate cell behaviour). Mapping of the differentiation landscape is well under way, evident from exponential accumulation of single-cell transcriptomics and functional data. However, the dissection of regulatory networks shaping the landscape and controlling cell fate is still in its infancy. Our study sets out a blueprint of how to tackle this problem. Using a model of lympho-myeloid progenitors, we demonstrate how interactions between network components can be established, thus enabling construction of functional networks of highly predictive value. Additionally, our DoT score method combined with a scRNA-Seq landscape reference transforms the

interpretation of gene expression data, well beyond what is possible using gene/category enrichment analysis. Our work therefore contributes to the major goal of defining regulatory networks so that they can be exploited for targeted modulation of cell behaviour, including directed differentiation and reprogramming for cell therapy approaches, as well as differentiation therapy to tackle a wide range of malignancies.

# Materials and Methods

### Cell lines and culture conditions

Hoxb8-FL cell line was kindly provided by the Hans Häcker laboratory. Cells were grown at 37°C/5% $CO_2$ in: DMEM (Sigma R8758) with the addition of 10% FCS (ES-culture compatible), 5% Flt3L conditioned medium, 50 μM 2-Mercaptoethanol, 1% Penicillin + Streptomycin solution (Sigma P0781), 1% Glutamine solution (200 mM stock solution—Sigma G7513), and 1 μM β-oestradiol (Sigma E2758). Cells were maintained at densities between $10^5$ and $1.5 \times 10^6$ cells/ml.

Conditioned medium was prepared using the B16 cell line constitutively expressing Flt3L, also provided by Hans Häcker. Cells were grown at 37°C/5% $CO_2$ in: DMEM (Sigma R8758) with addition of 10% FCS (ES-culture compatible), 50 μM 2-Mercaptoethanol, 1% Penicillin + Streptomycin solution (Sigma P0781), and 1% Glutamine solution (200 mM stock solution—Sigma G7513). To produce the conditioned medium cells were grown until confluent, supernatant was harvested and replaced daily over a 3-day period. The harvested supernatant was filtered, aliquoted and stored at -80°C for further use.

293T cells were grown at 37°C/5% $CO_2$ in DMEM (Sigma R8758) with addition of 10% FCS (Sigma F7524), 1% Penicillin + Streptomycin solution (Sigma P0781), and 1% Glutamine solution (200 mM stock solution—Sigma G7513). To prepare lentiviral supernatants, 293T cells were grown to 90% confluency in 10 cm dishes or 6-well plates (smaller scale preparation used for the TF screen) and transfected using the TransIT-LT1 (Mirus MIR2300) reagents following manufacturer's instructions with 5 μg (1.1 μg for the smaller scale) of each plasmid: sgRNA expression transfer plasmid (pBA439/GBC; Adamson *et al*, 2016; Dixit *et al*, 2016), pMD2G and ΔR8.9. Proceeding overnight culture 293T cells were switched to the culture medium (6 ml or 1.4 ml) of the cell line used for infection in subsequent experiments. Viral supernatant was harvested the following day, filtered through a 0.45-μm filter, aliquoted and stored −80°C for further use. Respective control and treatment lentiviral supernatants were prepared in parallel.

Hoxb8-FL cells expressing the Cas9 protein (Hoxb8-FL + Cas9) were generated by transduction with pKLV2-EF1a-Cas9Bsd-W (obtained from the Vassiliou Lab; Tzelepis *et al*, 2016) and subsequent selection in medium containing 10 μg/ml Blasticidin (Invivogen antbl-05) for 5 days. Prior to an experiment, cells were again selected in Blasticidin for 2-3 days and allowed to recover for 24–48 h before any other treatment.

Hoxb8-FL cells with Gata3 KO were generated by transduction with the lentiviral GBC library vector containing Gata3 sg2 sgRNA. 48 h after infection single cells were sorted into 96-well plates and cultured under normal conditions. After expansion (Gata3 sgRNA-transduced cell initially exhibited retarded growth), we inspected

the Gata3 genotype using high-throughput sequencing, with protocol analogous to the one described in section "Verification of CRISPR efficiency". For the ChIP-Seq analysis, we selected a clone carrying two alleles with frameshift mutations.

## Cloning

sgRNA sequences were derived from one of the following: the Brie library (Doench *et al*, 2016), Mouse v2 CRISPR library (Tzelepis *et al*, 2016), (Dixit *et al*, 2016) or (Gundry *et al*, 2016). List of sgRNA and oligonucleotide sequences are provided in Dataset EV1. Oligonucleotides carrying sgRNA sequences were cloned into the Perturb-Seq GBC library (Addgene 85968). Respective pairs of oligonucleotides were annealed in 50 μl annealing buffer (100 mM potassium acetate, 30 mM Hepes-KOH at ph 7.4, 4 mM magnesium acetate). Annealing mix was incubated at 95°C for 5 min and gradually cooled down to room temperature and stored at −20°C. For the ligation, 2 μl of annealed oligos (1:20 diluted) was mixed with 100 ng of the vector backbone (digested with BstXI and BlpI), 2 μl of 10× T4 ligase buffer and 1 μl of T4 ligase (M0202S) in a 20 μl reaction. Ligation was incubated at 16°C overnight and transformed into DH5α bacteria (NEB C2987P). Clones were isolated and verified by sequencing.

## CRISPR/Cas9 perturbation

To perturb TFs in Hoxb8-FL+Cas9 cells, we used the following protocol: $3.3 \times 10^5$ cells in 0.56 ml of media were seeded in a 24-well plate with 90 μl viral supernatant and 5.33 μl of polybrene (1 mg/ml stock solution, Sigma TR-1003-G), cells were centrifuged for 90 min at 780 *g*, at 32°C, incubated at 32°C for 1.5 h and cultured overnight at 37°C. The following day cells were washed to remove the viral particles. In the case of non-essential genes (non-dropout genes in Fig 2A and the Max gene), 0.44 ml of medium was added on day 1 and cells were split 1:2 into 12-well plates on day 2. Cells were harvested on day 4 and stained with 7AAD (BD 559925), and 375 sgRNA-expressing (BFP$^+$) cells were sorted into individual wells of a 96-well PCR lysis plate containing lysis buffer (see below). Plates were stored at −80°C for further processing. For essential genes (excluding Max, see above), 0.6 ml of medium was removed and 1.04 ml of fresh medium was added on day 1. On day 2, cells were harvested and sorted as above. To generate the main TF network, each TF was targeted by 3 sgRNAs, across three culture wells, and a total of 8 replicates were analysed. Each 96-well plate contained samples perturbing 3 different TFs and 16–24 samples of control cells, i.e. infected either with a non-targeting sgRNA (specific to GFP, absent in the genome) or targeting the Rosa26 locus. To analyse the consequences of removing Hoxb8 ectopic expression, cells were infected with the empty vector control as above but β-oestradiol was withdrawn from the culture for the last 18 h, and 8 biological samples were collected across three different cultures.

For double perturbation experiments, Hoxb8-FL + Cas9 cells were transduced (as described above) with lentiviral vectors encoding sgRNAs targeting indicated TFs or control sgRNAs. After 30 h, cells were washed with media and cultured for another 18 h either with or without β-oestradiol. Further processing was performed analogously, and scRNA-Seq libraries were generated as detailed in the section "Sample processing for scRNA-Seq" (with increased

RNase inhibitor concentration). Libraries were sequenced using the Illumina NovaSeq instrument, obtaining approx. 350 mln reads per 88 samples. Factors Cebpa, Meis1 and Spi1 were targeted by 3 sgRNAs each. In total, 24 replicates were analysed for: control cells, TF perturbed cells (3 sgRNAs, 8 replicates each), TF perturbed cells without β-oestradiol (3 sgRNAs, 8 replicates each) and 16 replicates for control cells without β-oestradiol.

For competitive cultures between TF perturbed and control cells, the Hoxb8-FL and Hoxb8FL + Cas9 cells were infected as described above. Each condition was performed in triplicate. Throughout the experiment, cells were cultured in a 24-well plate and split to maintain cell density below $10^6$ cells/ml. The 7AAD$^-$/BFP$^+$ fraction of cells was analysed for each sample on days: 2, 3, 5, 7, 9 and 11 using the BD LSRFortessa flow analyser. The relative fraction of BFP$^+$ cells was computed by normalising the BFP$^+$ fraction observed in Hoxb8-Cas9 cells to the mean BFP$^+$ fraction in respective Hoxb8-FL control cells, thus cancelling out differences in infection efficiencies.

## Verification of CRISPR efficiency

Hoxb8-FL + Cas9 cells were infected as described above using the sgRNA targeting the Ptprc (CD45) locus. Cells were cultured for 4 days as above before being stained and analysed for Ptprc protein levels and BFP$^+$ fractions on the BD LSRFortessa flow analyser. Cells were stained as follows: harvested cells were resuspended in 100 μl of 2% FCS/PBS solution and incubated with 1 μl of CD45-FITC antibody (clone: 30-F11, BioLegend 103107) for 30 min on ice, washed twice with 2 ml of 2% FCS/PBS solution and resuspended in 500 μl of 2% FCS/PBS with 7AAD (BD 559925).

To verify the fraction of loci successfully mutated across multiple sgRNA treatments, we applied a deep-sequencing-based strategy. Hoxb8-FL + Cas9 cells were infected with respective constructs as described above, cultured for 5 days under normal culture conditions, then selected in medium with 4 μg/ml Puromycin (Invivogen ant-pr-1) and $1–2 \times 10^6$ cells were frozen for further processing. Genomic DNA was isolated using the QIAGEN AllPrep DNA/RNA mini kit (Qiagen 80204) according to the manufacturer's instructions. Loci around each of the sgRNA-targeted sites were amplified by PCR using primers listed in Table EV1, adding sequences complementary to the Illumina adapters. One sample (JunB, sgRNA2) failed to amplify and was omitted from the analysis. Illumina indices were added in a 2$^{nd}$ round of PCR using the Illumina Nextera XT Index Kit v2. Libraries were purified using the AMPure XP beads (Beckman A63882) and sequenced on a Mi-Seq machine, using the MiSeq Reagent Nano Kit v2 (Illumina MS-103-1001). 5,000 reads were analysed and aligned against the reference sequence, subset for reads matching the leader sequence and sufficient number of matches. For each sample, a fraction of reads with frameshift mutations (insertion or deletions) was calculated.

## Sample processing for RNA-Seq

For the main TF screen, samples were processed using a modified version of the Smart-Seq2 protocol (Picelli *et al*, 2014; Bagnoli *et al*, 2018) described below. 375 cells were sorted into 11.5 μl lysis buffer containing 0.575 μl of SUPERase-In RNase Inhibitor (20 U/μl Thermo Fisher AM2694) and 0.23 μl of 10% Triton X-100 solution

(Sigma 93443), vortexed and stored at −80°C. After thawing on ice, 5 μl of annealing solution (0.5 μl of ERCC RNA Spike-In Mix (1:300,000 dilution, Thermo Fisher 4456740), 0.1 μl of the oligo-dT primer (100 μM stock concentration)) was added. Samples were incubated at 72°C for 3 min and cooled down on ice, and 1/5 of the volume was used for further processing. The reverse transcription was performed by: adding 0.1 μl Maxima H Minus enzyme (200 U/μl, Thermo Fisher EP0752), 0.25 μl of SUPERase-In RNase Inhibitor, 2 μl of Maxima RT buffer, 0.2 μl of the TSO oligo (100 μM stock concentration), 1.875 μl of PEG 8000 (Sigma P1458) and 1 μl of dNTPs (10 mM stock concentration Thermo Fisher 10319879) to a total volume of 10 μl followed by 90-min incubation at 42°C and 15-min incubation at 70°C. cDNA was amplified by adding 1 μl of the Terra PCR Direct Polymerase (1.25 U/μl, Takara 639270), 25 μl of the Terra PCR Direct buffer and 1 μl of the ISPCR primer (10 μM stock concentration) to a total volume of 50 μl, and PCR conditions were as follows: 98°C for 3 min, 98°C for 15 s, 65°C for 30 s, 68°C for 4 min (13 cycles) and 72°C for 10 min. The PCR product was purified using AMPure XP beads (Beckman A63882). Remaining steps were carried out according to the standard Smart-Seq2 protocol. For Plates 16–18, the concentration of the RNAse inhibitor was doubled in the lysis buffer and 0.115 μl of 100 mM DTT concentration was added. Libraries were sequenced using the Illumina Hiseq4000 instrument, obtaining 350–400 mln reads per 96 samples.

In the case of the pilot experiment (Experiment 1 in Fig 1E and time-course data in Fig EV1E), the samples were processed analogously with one difference: 75 cells were sorted into 2.3 μl lysis buffer, and the entire solution was processed as above.

## ChIP-Seq

$10^8$ Hoxb8-FL + Cas9 cells were harvested and fixed in 1% formaldehyde for 10 min at room temperature. Reaction was quenched by adding 0.125 M glycine and incubated for 5 min at room temperature. Cells were washed in ice-cold 1× PBS, resuspended in cell lysis buffer (10 mM Tris pH 8.0, 10 mM NaCl and 0.2% NP40) containing protease inhibitors (leupeptin, NaBu and PMSF) and incubated on ice for 10 min. The nuclei were collected by centrifugation at 600 g for 5 min. at 4°C, resuspended in nuclei lysis buffer (50 mM Tris pH 8.0, 10 mM EDTA, 1% SDS) with protease inhibitors (leupeptin, NaBu and PMSF) and incubated on ice for 10 min. One ml of IP dilution buffer (20 mM Tris pH 8.0, 2 mM EDTA, 150 mM NaCl, 1% Triton X-100, 0.01% SDS) with protease inhibitors (leupeptin, NaBu and PMSF) was added, and chromatin was sonicated at 4°C in a Bioruptor (Diagenode) with: 5–7 cycles (30 s on and 30 s off). The fragmented chromatin was centrifuged at 3,220 g for 10 min, and supernatant after transferring was diluted 4 × with IP buffer. The chromatin was pre-cleared as follows: 25 μl of rabbit IgG (2 μg/μl, Sigma I5006) was added and incubated at 4°C for 1 h, 200 μl of Protein G sepharose beads (Roche, 1:1 slurry in IP dilution buffer) was added and incubated at 4°C for 2 h, and beads were harvested at 1,791 g for 2 min at 4°C. Samples were subsequently incubated with the respective antibodies at 4°C for overnight with rotation, and 60 μl of protein G agarose beads (1:1 slurry in IP dilution buffer) was added and incubated with the samples for 2 h with rotation. Beads were collected by centrifugation at 5,400 g for 2 min and washed twice with low salt buffer (Tris pH 8.0, 2 mM EDTA, 50 mM NaCl, 1% Triton X-100,

0.1% SDS), once with LiCl buffer (10 mM Tris pH 8.0, 1 mM EDTA, 0.25 M LiCl, 1% NP40, 1% Sodium deoxycholate monohydrate) and twice with TE pH 8.0. Complexes were eluted twice by adding 150 μl elution buffer (100 mM NaHCO$_3$, 1% SDS). Cross-linking was reversed by addition of 0.3 M NaCl, and RNA was digested with RNase-A during an overnight incubation at 65°C. Samples were treated with Proteinase K for 2 h at 45°C, and DNA was purified using Qiagen PCR clean up columns. Illumina libraries were prepared using the Illumina TruSeq DNA Sample Prep Kit (Illumina IP-202-1012), size selected by gel purification (250–450 bp) and sequenced using Illumina HiSeq 2500 or HiSeq4000 instruments

Antibodies used: CEBPα (Santa Cruz sc-61x), CEBPβ (Santa Cruz sc-150x), Gata3 (Cell Signalling D13C9), Ebf1 (Millipore ABE1294), Tcf3 (Santa Cruz sc-763), Tal1 (Santa Cruz sc12984x), Meis1 (Santa Cruz sc-10599x), Spi1 (Santa Cruz sc-352x), Runx1 (Abcam, ab23980-100), Erg1 (Santa Cruz sc354x), Lmo2 (R&D AF2726), Fli1 (Abcam ab15289-500), Lyl1 (Abcam ab15289-500), Gfi1 (Abcam ab21061), Gfi1b Santa Cruz sc8559x) and H3K27Ac (Abcam ab4729). The anti-Gata3 antibody specificity was confirmed using a Gata3 KO cell line (Appendix Fig S6), and the remaining antibodies have been used in previous works (Treiber *et al*, 2010; Wilson *et al*, 2016).

## ChIP-Seq data analysis

Sequencing data were pre-processed as previously described (Sanchez-Castillo *et al*, 2015). We used the IgG control as the background for all samples, except the Gata3, where we used data from Gata3-ChIP performed in the Gata3 KO cell line. For each sample, peaks were called using MACS with *P*-value cutoff of $10^{-5}$; if the number of peaks exceeded 7,000, then top 7,000 peaks with lowest *P*-values were used. Downstream analysis was performed using R language (R Core Team) and indicated packages. Peaks overlapping blacklisted regions were removed (Amemiya *et al*, 2019). Replicates were subset to a common set of overlapping peaks. ChIP-Seq experiments for Fli1, Erg1, Runx1, Lmo2, Tal1, Gfi1, Gfi1b, Meis1, Spi1 and Cebpb were performed as single replicates; Tcf3, Ebf1 and Gata3 were performed in triplicate and Cebpa in quadruplicate. One sample for Tcf3 showed poor signal and was excluded from the analysis. Peaks were mapped to genes accordingly: peaks overlapping regions 1,000 bp upstream to 200 bp downstream of a TSS were mapped to the corresponding gene, and peaks overlapping a gene body were mapped to the respective gene and intergenic peaks within 50 kb of the nearest genes were assigned to the first closest gene. Expected values and *P*-values for overlaps between genes differentially expressed and genes with mapped peaks were calculated using matching hypergeometric distributions. Gene annotations were extracted from the TxDb.Mmusculus.UCSC.mm10.knownGene package, and genomic features were annotated using the annotatePeak function from the ChIPseeker package. Data are available on GEO with accession number: GSE146128 and CODEX (http://codex.stemcells.cam.ac.uk/) databases. An interactive UCSC browser session is available at: http://genome-euro.ucsc.edu/s/idk25/TFnet2020_allChIPs_impr.

To generate Fig EV4C, previously published Ebf1 ChIP-Seq datasets were retrieved from Cistrome DB (Mei *et al*, 2017). Peak coclustering was essentially performed as previously described (Edginton-White *et al*, 2019). Briefly, an intersection matrix for all

combinations of intersections between peak sets was computed using *pybedtools intersection_matrix* (Dale *et al*, 2011). A Sørensen–Dice coefficient (Dice, 1945; Sorensen, 1948) was computed for each intersection relative to the parent peak sample sizes.

### Digital genomic footprinting analyses

Digital genomic footprinting was carried out on previously published ATAC-Seq date generated from Hoxb8-FL cells (Basilico *et al*, 2020) using the dnase_footprints function from the pyDNase package (Piper *et al*, 2013) with the -A parameter. Motif discovery and annotation to footprints were performed using the findMotifsGenome and annotatePeaks functions (Homer package), respectively (Heinz *et al*, 2010). Venn diagram overlaps of footprints were obtained using ChIPpeakAnno (Zhu *et al*, 2010). Footprinting matrices were created using dnase_to_javatreview (pyDNase package), with heatmaps generated using Java TreeView (Saldanha, 2004). To correlate with gene expression data, annotation to the nearest gene was performed using bedtools closest function (Quinlan & Hall, 2010) with the -t first parameter. Ebf1 KO or Gata3 KO versus wild-type gene expression changes ($log_2$(fold change)) were retrieved for genes with Ebf1 and Gata3 footprints, respectively. As controls, random regions of similar sample and sizes were computed using bedtools random and underwent the same treatment as footprints.

### Statistical analysis

RNA-Seq data were modelled using a negative binomial distribution based on the DESeq2 methodology (Love *et al*, 2014), using adjusted *P*-values (Benjamini–Hochberg method) followed by the indicated $log_2$ (fold change) filters to call differential expression. For the single perturbation experiments, we built models with one coefficient corresponding to the perturbation effect and a blocking coefficient corresponding to the fraction of reads mapped to the intronic regions per sample. For the double perturbation experiment, models included a coefficient for each perturbation effect, the interaction term and the blocking coefficient. Statistical tests for enrichment overlap or enrichment were performed using hypergeometric distribution with matching parameters (base R language (R Core Team)). Data correlations shown in scatter plots were analysed with a linear model indicated by the equation, and shaded areas indicate confidence intervals for the fit (R language, ggplot2 package).

### RNA-Seq and network analysis

Sequencing reads were aligned to the mouse genome (mm10) or human genome (hg19) using the STAR aligner (version 2.7.3a) with default parameters. Reads mapping to exons and introns were counted separately with featureCounts (version 2.0.0) using the ENSEMBL v93 annotation. Introns were defined as all regions in between the exon ranges within each gene. Genome sequence and annotation were augmented with details of ERCC RNAs, GBC library backbone and Cas9-Bsd expression plasmid where applicable.

For comparison of gene expression values across datasets (Fig 6A and B, and EV4G), we used data available from GEO (GSE127267, GSE107240; Revilla-i-Domingo *et al*, 2012; Li *et al*,

2018), processed as above and normalised using DESeq2 package (Love *et al*, 2014).

Each sample in the TF screen was subjected to a quality control, samples with: < 500,000 reads, < 30% of reads mapped to exons, > 12% of reads mapped to ERCC transcripts, > 5 % mitochondrial reads or < 4,000 genes detected above 10 counts per million were discarded. 1,138 out of 1,148 samples passed quality control. We analysed samples from each plate using principal component analysis (PCA) and observed that a considerable part of variation was correlated with the fraction of reads mapped to intronic regions, which is inversely correlated with the fraction of reads mapped to the exons. We believe that this may be due to either differences in RNA quality or degree of isolated nuclear RNA (nuclei lysis). As the fraction of intronic reads was evenly and randomly distributed among control and treated sample, we could effectively remove this effect by either linear regression (for downstream estimation of gene expression values) or including fraction of intronic reads as a covariate in the differential expression model. Inspection of the PCA analysis highlighted 7 clear outlier samples, which were removed from further analysis.

As a basis for edges in our network, we used differential expression. We performed the analysis using the DESeq2 software (Love *et al*, 2014), subsetting for samples in the same plate and including fraction of reads mapped to intronic regions in each sample as a covariate. We first compared samples treated with each sgRNA separately and found that the observed changes are highly correlated in almost all cases; thus to construct the full network, we compared all sgRNA samples with all the controls. In the case of Meis1, Rad21 and Max, one of each sgRNAs had a much weaker effect on the gene expression and were therefore excluded from the further analysis. The remaining sgRNAs showed strong and correlating effects. To minimise batch effects due to common control samples in each plate, we also tested differential expression against the assembly of all controls and included only genes passing both tests. Although some within-plate correlation remains visible in cases of samples with small numbers of detected DE, we cannot exclude that it corresponds to real signal, e.g. targets of Lmo2 and Ldb1 in Plate7 correlate and indeed are expected to share a large number of targets based on their function. For selection of targets, we applied thresholds of adjusted *P*-value 0.1 and minimal |$log_2$(fold change)| > 0.2.

For the double perturbation experiments, we used the DESeq2 software to fit a two-factor model with interaction and blocking for the fraction of reads mapped to intronic regions (model formula: ~perturbation1*perturbation2 + intron_fraction). To define differentially expressed genes, we considered genes with coefficients passing thresholds of adjusted *P*-value < 0.1 and |$log_2$(fold change)| > 0.2. We classified interactions either at: (i) low stringency, selecting only genes DE for both single perturbations and filtering the interaction changes with |$log_2$(fold change)| > 0.2 criterium; and (ii) high stringency, considering only genes with significant interaction terms (adjusted *P*-value < 0.1 and |$log_2$(fold change)| > 0.2) and classifying the other two coefficients using the same criteria.

The network was visualised using Gephi (0.9.2) (Bastian *et al*, 2009) with Force Atlas 2 algorithm. The relations between TFs was summarised as follows: Fig 3A and B—correlation of observed $log_2$(fold changes) among common targets between each pair of

TFs, Appendix Fig S3A—number of shared gene targets and z-score value computed based on the matching hypergeometric distribution and Appendix Fig S3B—correlation of observed $\log_2$(fold changes) shrunk using the adaptive shrinkage estimator (Stephens, 2017) computed on all genes expressed in Hoxb8-FL cells.

Target gene modules were identified using TFs with > 200 target genes identified, hierarchical clustering of observed $\log_2$(fold change) values (shrunk with the adaptive shrinkage estimator) using (1-correlation) as distance measure and average linkage method. The resulting tree was cut dynamically with min. cluster size of 40 and cutHeight of 0.4 (dynamicTreeCut package). This results in a single large cluster containing the majority of genes with no clear regulatory patterns and 46 smaller clusters exhibiting relevant patterns. Enrichment analysis was performed using the Enrichr software (Kuleshov et al, 2016).

Computer code used for the analysis is available in the repository: https://github.com/Iwo-K/TFnet2020.

### scRNA-Seq data analysis

Data analysis was performed using functions available in the Scanpy package (Wolf et al, 2018). For human bone marrow mononuclear cells, the matrix of cell x genes counts was obtained from Human Cell Atlas (https://data.humancellatlas.org/explore/projects/cc95ff 89-2e68-4a08-a234-480eca21ce79). To perform quality control, cells with < 600 Genes detected and > 10% of counts mapped to mitochondrial genes were excluded. Data for remaining 235,735 cells were log-normalised, and 8,498 highly variable genes were identified. After scaling, the expression values of the highly variable genes were used to compute 50 principal components, and these were used to identify 15 nearest neighbours and compute clusters (Leiden algorithm) (Traag et al, 2019) and the UMAP embedding (Preprint: McInnes et al, 2018). Data annotation was performed manually using known marker genes for each lineage, e.g. CD34 for early progenitors, PF4 for megakaryocytes, CD3E for T cells, CD19 for B cells, DNTT/IL7R for lymphoid cells, IRF8 for monocytes/dendritic cells, ELANE/PRTN3 for neutrophils, KLF1/GATA1 for erythroid cells and MS4A2 for basophils and mast cells.

For the mouse LK + LSK landscape, the matrix of cell x genes counts was obtained from (Dahlin et al, 2018). Data were analysed and annotated as above, using 5,140 highly variable genes (excluding 368 genes associated with cell cycle), 50 principal components and 5 nearest neighbours to compute the clustering and UMAP embedding.

In the case of human foetal liver data, the matrix of cell x genes counts was obtained from ArrayExpress (E-MTAB-7407; Popescu et al, 2019). Cells expressing < 1,000 genes and with > 10% mitochondrial gene counts were excluded. Data were subset for blood and endothelial populations according to the annotation provided by the original authors and processed as described above to compute 7,984 highly variable genes and 50 principal components. To integrate the numerous batches of data, we excluded batches with < 400 cells and used batch balanced k-nearest neighbour method (3 neighbours in each batch) (Polanski et al, 2020) to learn a common neighbour graph. To isolate the HSPC and lymphoid cell populations, we clustered the data with the Leiden algorithm and subset for the relevant populations. The data were re-processed as above, using 8,886 highly variable genes, 30 principal components and batches with

$\geq$ 100 cells, and cells belonging to the HSPC-preB trajectory were isolated (excluding mature B cells, T-cell progenitors, NK cells and ILC precursors). Pseudotime values were computed using the diffusion pseudotime method (Haghverdi et al, 2016). Expression values were averaged using a sliding window (size of 400 cells and step size of 100) followed by a loess curve fit.

Mouse LMPP/ALP/BLP data (WT cells only) were obtained from GEO (GSE101735; Loughran et al, 2017), aligned and counted as described in the "RNA-Seq and network analysis" section. To remove low-quality cells, we excluded cells with < 25% reads mapped to exons, < 100,000 counts, > 12% of reads mapped to ERCC RNAs, < 2,000 genes detected at 50 counts per million and > 15% of reads mapped to mitochondrial genes. Data were processed as above using 7,000 highly variable genes, 50 principal components, 7 nearest neighbours and excluding 368 genes associated with cell cycle. Cells were arranged according to the pseudotime values (diffusion pseudotime), and their expression values were averaged using a sliding window (size of 50 cells and step size of 20 cells) and a loess fit. Cell type annotation is based on the isolated populations using flow cytometry and was provided by the original authors (Loughran et al, 2017).

### Cell projection across landscapes

To find the most similar cells between datasets (Figs 1C and 7A–C), we identified the nearest neighbours between them. Briefly, target and reference datasets were jointly normalised and scaled. Principal components were computed using the reference dataset, and the target data were projected onto the PCA space (pca.transform function from the sklearn module), and nearest neighbours were computed based on the pairwise Euclidean distances between samples. Cells in the reference landscape (e.g. landscape in Fig 1C) were colour-coded based on the total number of nearest neighbours identified in the target dataset (projection score), reflecting relative transcriptional similarity.

### DoT score

DoT scores for all cells (vector **s**) are defined as: $\mathbf{s} = \mathbf{Xv}$, where $\mathbf{X}$ is a matrix cells x genes with scaled expression values and $\mathbf{v}$ is a vector of weights (in our case value of $\log_2$(fold change) for each gene). This is equivalent to computing the dot product of the vector of scaled expression values, and the vector of weights for each cell, i.e., is proportional to the cosine of the angle between the two vectors. In order to normalise values across different experiments and assign a statistical significance, we calculated a z-score for each cell against a simulated DoT score distribution. We ran 500–1,000 simulation randomly assigning weights from the set of all observed $\log_2$(fold changes) and all expressed genes in the Hoxb8-FL cells, with the number of non-0 weights equivalent to the original weights vector (here: the number of DE genes observed for the specific perturbation). As scaling of the reference landscape is important for the interpretation of the DoT score (it provides a "starting point"), we used mean expression values best corresponding to the Hoxb8-FL transcriptome. For LK+LSK mouse data, we used cell projection data (see section above) to choose the most appropriate cluster. For the human BMMC data, we chose the most immature cluster, containing CD34[+] cells. The relevant code is available as a python module - https://github.com/Iwo-K/dotscore.

## Data availability

- RNA-Seq data: Gene Expression Omnibus GSE146128.
- ChIP-Seq data: Gene Expression Omnibus GSE146128 and CODEX (http://codex.stemcells.cam.ac.uk/).
- Interactive genomic browser session (ChIP-Seq data): http://genome-euro.ucsc.edu/s/idk25/TFnet2020_allChIPs_impr.
- DoT score computer code: https://github.com/Iwo-K/dotscore.
- Analysis computer code: https://github.com/Iwo-K/TFnet2020.

**Expanded View** for this article is available online.

## Acknowledgements

Work in the Gottgens Laboratory is funded by grants from Wellcome (206328/Z/17/Z); MRC (MR/S036113/1); Blood Cancer UK (18002); Cancer Research UK (C1163/A21762); National Institutes of Health (NIDDK DK106766); and core support grants by the Cancer Research UK Cambridge Centre (C49940/A25117); and by Wellcome and MRC to the Cambridge Stem Cell Institute (203151/Z/16/Z). The authors thank Reiner Schulte, Chiara Cossetti, Gabriela Grondys-Kotarba and Annie Hoxhalli from the Cambridge Institute for Medical Research Flow Cytometry Core facility for their assistance with cell sorting. We would also like to thank the Cancer Research UK Cambridge Institute Genomics Core Facility for performing high-throughput sequencing.

## Author contributions

IK performed the TF network experiments and analysed the data. IK and NKW generated ChIP-Seq data. SJK generated sgRNA expression constructs. RH provided help with ChIP-Seq data analysis and data deposition in public repositories. PC, AL and RG contributed to the Ebf1 function analysis and helped with the manuscript preparation. IK and BG conceived and designed the study. BG supervised the study. IK, NKW and BG wrote the manuscript.

## Conflict of interest

The authors declare that they have no conflict of interest.

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
