## [Review Process File · The EMBO Journal]

Interactions between lineage-associated transcription factors govern haematopoietic progenitor states

Iwo Kucinski, Nicola Wilson, Rebecca Hannah, Sarah Kinston, Pierre Cauchy, Aurelie Lenaerts, Rudolf Grosschedl, and Berthold Gottgens

DOI: [10.15252/embj.2020104983](https://doi.org/10.15252/embj.2020104983)

Corresponding authors: Berthold Gottgens (bg200@cam.ac.uk)

Review Timeline:

Submission Date:	12th Mar 20
Editorial Decision:	6th May 20
Revision Received:	31st Jul 20
Editorial Decision:	1st Sep 20
Revision Received:	7th Sep 20
Accepted:	8th Sep 20

Editor: Daniel Klimmeck

Transaction Report:

Dear Bertie,

Thank you for the submission of your manuscript (EMBOJ-2020-104347) to The EMBO Journal. Please accept my sincere apologies for the very unusual delay with the peer-review of your manuscript due to protracted referee input during the last weeks. Your manuscript has been sent to three reviewers but two of them got significantly delayed as to other obligations during the pandemic. We have now received reports from all three referees, which I enclose below.

As you will see, the referees acknowledge the novelty and robustness of your results, although they also express a number of issues that will have to be conclusively addressed before they can be supportive of publication of your manuscript in The EMBO Journal. In more detail, referee #2 raises concerns regarding the claims made on cooperativity between the TFs analysed (ref#2, pt.1). Further, referee #2 states that the hierarchy between primary and secondary responses should be better addressed (ref#2, pt.2). In addition, the reviewers raise a number of points related to minor complementary analyses, data presentation, improved discussion of the findings as well as wording, which would need to be conclusively addressed to achieve the level of robustness and clarity needed for The EMBO Journal.

I judge the comments of the referees to be generally reasonable and given their overall interest, we are in principle happy to invite you to revise your manuscript experimentally to address the referees' comments.

I would appreciate if you could contact me during the next weeks via e.g. a video call to discuss your perspective on the comments and plan for revisions. Also, please let me know if you have additional questions or need further input on the referee comments.

Please see below for additional instructions for preparing your revised manuscript.

In this context I also want to point to our adjusted GTA. We are aware that many laboratories cannot function at full efficiency during the current COVID-19/SARS-CoV-2 pandemic and have therefore extended our 'scooping protection policy' to cover the period required for a full revision to address the experimental issues highlighted in the editorial decision letter. Please contact us at any time to discuss an adapted revision plan for your manuscript should you need additional time.

Thank you for the opportunity to consider your work for publication. I look forward to your revision.

Best regards,

Daniel

Daniel Klimmeck, PhD
Editor
The EMBO Journal

Before submitting your revision, primary datasets (and computer code, where appropriate) produced in this study need to be deposited in an appropriate public database (see <https://www.embopress.org/page/journal/14602075/authorguide#datadeposition>).

The accession numbers and database should be listed in a formal "Data Availability" section (placed after Materials & Method) that follows the model below (see also <https://www.embopress.org/page/journal/14602075/authorguide#availabilityofpublishedmaterial>). Please note that the Data Availability Section is restricted to new primary data that are part of this study.

Data availability

Our journal also encourages inclusion of *data citations in the reference list* to directly cite datasets that were re-used and obtained from public databases. Data citations in the article text are distinct from normal bibliographical citations and should directly link to the database records from which the data can be accessed. In the main text, data citations are formatted as follows:

"Data ref: Smith et al, 2001" or "Data ref: NCBI Sequence Read Archive PRJNA342805, 2017". In the Reference list, data citations must be labeled with "[DATASET]". A data reference must provide the database name, accession number/identifiers and a resolvable link to the landing page from which the data can be accessed at the end of the reference. Further instructions are available at <https://www.embopress.org/page/journal/14602075/authorguide#referencesformat>

- a point-by-point response to the referees' comments, with a detailed description of the changes made (as a word file).
- a word file of the manuscript text.
- individual production quality figure files (one file per figure)
- a complete author checklist, which you can download from our author guidelines (<http://emboj.embopress.org/authorguide>).
- Expanded View files (replacing Supplementary Information)

Further information is available in our Guide For Authors:

The revision must be submitted online within 90 days; please click on the link below to submit the revision online before 4th Aug 2020.

Link Not Available

Referee #1:

In this study, Kucinsky et al. use of a fetal liver derived hematopoietic progenitor cell line in studying the role of 39 transcription factors with previously reported hematopoietic functions, using a knockout approach. Cells targeted for each of the individual factors were analyzed by single cell RNAseq. The results provided a complex picture of hematopoietic transcription factor networks active in cell fate decisions, some of which were known and others, such as the Ebf1-Gata3 synergism associated with the induction of a myeloid expression program, were unexpected and novel. The authors furthermore developed a new method (DoT) to visualize and predict cell fate transitions as a consequence of knocking out specific factors.

Besides delivering significant new insights into hematopoietic differentiation, the data generated in this study represent a valuable resource for investigators in the field and the DoT method should become a useful tool that can be applied for other types of single cell sequencing analyses. Overall the work is of high quality and the data well presented.

Comments and criticisms

My only major problem with the paper is that it is very hard to read, which is in part based on the complexity of the problem and the computational analyses performed, some of which will not be familiar to most readers.

Some of the points below list specific items that relate to this issue.

1. Of the 39 genes that were perturbed, Rad21 is not known to encode a TF and this reviewer has not heard of noB. Please explain
2. Fig1C. What do the yellow and blue dots represent? Why are there only so few?
3. FigS1F. The Venn diagram shows a surprisingly small degree of overlap between the 3 sgRNAs tested. Please comment.
4. Fig. 2B. This diagram is hard to understand. Specific interactions of factors depicted in the center (Myb, Cebpa, Runx1, etc.) are not well visualized. To improve this, additional representations of this graph would be useful to be shown in the supplement, such as blow-ups of the some areas with the most relevant factors discussed later in the text. Also, the yellow circular symbols indicating perturbed TFs are very hard to see.
5. The text does not specifically mention Figures 2A and B
6. Fig. 3B and C. I find it very difficult to understand these figures. Why are only subsets of TFs represented in the two figures? Why does activation of a target in C not always show a positive correlation in B (e.g., Myc-Mitf)? Why are no downstream TF targets shown for e.g. C/EBPa, for which PU.1 has been described as a partner?
7. Fig.3D is another one that is hard to understand. The clusters selected do not seem to represent the major ones in the scheme. What are all the other clusters?
8. Line 219 '..we also generated chromatin immunoprecipitation (ChIP-Seq) datasets for 14 TFs as well as the H3K27Ac histone modification..' To judge the quality of the data please show representative examples in the Supplement
9. Line 291: 'Ebf1 counterbalances cell growth and cell cycle. How exactly are these distinguished?
10. Line 428: 'The network presented here also shows that Cebpa expression promotes the cell growth programme' (Figure 3D). How does this fit with the literature showing that C/EBPa inhibits growth?
11. Fig.4. ChIPseq using what cells?
12. Fig. 5C. Please mention in the legend what is shown on the left and on the right.
13. Show a reference image of the human single cell landscape with areas denoting different cell types.

Referee #2:

Through the development of a combined CRISPR/Cas9 - RNAseq approach, the manuscript entitled "Cooperation of Lineage-associated Transcription Factors governs Haematopoietic Progenitor states" by Kucinski et al., aims to (a) investigate Transcription Factor (TF) -target gene dependencies, (b) the crosstalk between different TFs as well as (c) identify common regulatory pathways in hematopoietic progenitors. Furthermore, the authors developed a novel method (Direction of Transition: DoT) to estimate transcriptional changes in response to an insult (in this case KO of specific TFs).

HoxB8-FL cells, established in the Häkel lab and resembling LMPP in vitro, were chosen as a model system. Here, 39 different TFs, previously reported to be important in hematopoietic progenitors, were deleted using CRISPR/Cas9 technology. In order to address the functional connection between TF and target genes, only TFs causing the deregulation of more than 200 genes were subsequently followed up (19 out of 39). Integrating transcriptomic data after TFs KO, the authors were able to identify genes responding exclusively to a single TF, as well as genes dependent to two or more TFs. Furthermore, the authors identified also TF-TF dependencies elucidating not only crosstalk and co-regulations already known, but also novel interactions and novel roles (i.e. Ebf1). Finally, the authors used the plethora of data collected in this paper to develop a novel method to estimate phenotypical changes induced by specific alterations (in this case after TF KO).

The study is very stimulating and will help to shed light on the complex networks regulating hematopoietic progenitor states. The data and their analysis are of high quality. Furthermore, the study will be an inspiration for further research that use similar approaches to dissect regulatory networks along hematopoietic differentiation trajectories. The paper is very well written and it is easy to follow even though more technical and complex concepts are introduced. In general, the paper is highly interesting with significant novelty, which provides important new perspectives to the field.

Nevertheless, to further improve the study some issues as well as minor concerns should be addressed before publication:

1. Despite the impressive amount of data collected and the deep analysis performed, functional validation experiments are missing. Furthermore, since single TFs are deleted one at the time and not simultaneously, the paper addresses neither cooperativity nor synergism between the different TFs, despite that fact that this is frequently claimed in the text (i.e. line 147, 153, 159, 175, 181). Indeed, the experimental design and the analysis shown in the paper can highlight correlation or anti-correlation of the transcriptional response after single TF KO, without the possibility to finely dissect the regulatory networks governing cell identity. Nevertheless, addressing cooperation or antagonism between the different TFs analyzed will be of great interest and will give additional strength to the current version of the paper. In this light, selected multiple TFs could be simultaneously deleted and the transcriptional response should be analyzed and compared to the changes induced by the deletion of each single TF.

2. Based on previous work from the lab, the authors divided the TFs in "essential" or "not essential". Based on this categorization, two different time points were chosen to perform transcriptomic analysis: 2 days after deletion for essential TFs, 4 days after deletion for non-essential TFs. Unfortunately, this experimental setting is sub-optimal since the time points chosen are relative late ones affecting the downstream interpretation of the data. Indeed, both primary and secondary targets will be identified in the analysis (as also stated in the text: line 231) with no possibility of distinguish between the two. Furthermore, after 2 days from the deletion of essential TFs (i.e. Myc), cells are already massively dying (as also shown in fig. S1G) adding an additional confounding factor to the analysis. This ambiguity could be the explanation of the massive transcriptional changes

observed for some of the TFs, as well as for the numerous co-regulated targets identified. As also mentioned in the text, it is also a possible explanation for the poor correlation observed between transcriptional changes and binding of specific TFs to regulatory regions of deregulated genes. In order to distinguish between primary and secondary response and to give a more faithful screenshot of the functional interactions between TFs and target genes earlier time points should be included (at least for some of the candidates) and compared to the ones used in the current version of the manuscript.

3. The authors used CD45 locus as positive control to check deletion efficiency. While in the text it is claimed that 48% of the cells showed a successful CD45 inactivation, the FACS plot shown in Fig. S1A shows only 7% of the cells are BFP+ and have lost CD45 expression. The authors should explain this discrepancy.

4. Even though the efficiency of targeting was proven using CD45 as positive control, as well as showing the fraction of genomic DNA reads with frameshifts for a selected number of TFs, either gene expression or protein analysis should be provided by the authors, at least for some prominent examples, to demonstrate efficient reduction of mRNA and/or protein of all the TFs used in the study.

5. The authors suggest a novel mechanism of action for Ebf1 working in concert with Gata3. Nonetheless, due to the failure to discriminate between primary and secondary response, and since the authors showed that Gata3 is actually an Ebf1 target, it is reasonable to hypothesize that most of the shared target genes are mostly Gata3 ones. In order to address this point, transcriptomic analyses using shorter time point, as well as integration analysis between RNAseq and ChIPseq should be performed in order to identify Ebf1 unique targets along with Ebf1/Gata3 shared ones.

6. In order to identify groups of genes similarly deregulated by different TFs, the authors subdivide the transcriptomic changes into 47 modules. In the text, few key genes belonging to crucial molecular pathways were mentioned, yet, a global pathway enrichment analysis is missing. In order to have a clear overview of genes contained in each module, authors should perform pathway enrichment analysis (IPA or others).

7. Supplementary tables provided are not correct: they show data not related to the manuscript submitted.

8. Network analysis shown in Fig.3C does not highlight CEBP β regulation mediated by Spi1, as instead mentioned in the text (line 173). Please elaborate better or specify in the text which is the corresponding figure.

9. Often, along the text, deregulation of different genes is mentioned (i.e. line 175, 176, 193, among others). Please provide plots where expression levels of the mentioned genes are shown.

Referee #3:

The MS presents new data on the genetic basis of hematopoietic progenitor cell differentiation obtained by systematic KO of 39 TFs in a cell line model for multipotent progenitors (Hoxb8-FL). Data from ChIP-seq and single-cell RNA-seq experiments of these cells and ex vivo isolated hematopoietic stem and progenitor cells are integrated with this new data set. Also, a bioinformatic method, Direction of Transition, is introduced that aims at predicting how cell shift in the transcriptional landscape upon TF knockdown (or any other type of perturbation).

The new data obtained upon TF KO are a very important resource for the field, as the authors already demonstrate with several new mechanistic insights. These include the co-expression of Ebf1 and Gata3 (mediated at least in part by Ebf1 activating Gata3 transcription) in lymphoid-primed progenitors prior to committing to B cell development.

Overall, the MS is very well written and the results are described clearly. Edits or additions at a few places will improve clarity.

- Direct versus indirect TF effects (p.6: "... much of this uncertainty can be disentangled"): Could the authors be more specific here? Does knowing the network of TF interactions help decide whether the KO effect of one of those TFs on another target gene is direct or indirect? Can this be quantified?

- Fig. 3C: Reduce font size so as not to obscure arrowheads in a few places

- Fig. 4C,D: Improve annotation. Where is the +37kB enhancer for Cebpa in C? What does the red arrow point to in D?

- Fig. 6: Please explain A in some detail (in text or figure legend). Is there a viewpoint region in the landscape?

Some questions concerning DoT:

- It is not clear whether Euclidian distance is the right metric. If the transcriptional landscape is a nonlinear manifold in gene-expression space, then diffusion distance may be more appropriate. Please discuss.

- How does the direction vector field with respect to a viewpoint compare with RNA velocity?

Response to Editor's and Reviewers' comments

We would like to thank the editor and the reviewers for assessing our manuscript and appreciate the opportunity to improve our work. Below we outline how the main issues raised by the editor have been addressed, followed by a point-by-point to all the reviewers' comments:

Editor Comments:

1. In more detail, referee #2 raises concerns regarding the claims made on cooperativity between the TFs analysed (ref#2, pt.1)

1. We recognise that the experiments investigating TF cooperation through multiplex knock-out are of interest. However, we would also like to point out that in practical terms, this is not at all straightforward. Co-infecting cells with sgRNAs targeting two genes will generate a messy scenario, with approximately 50% of the cells not edited at all, 25% edited for just 1 gene, and only 25%% being true double knock-outs. Carrying out RNA-Seq on such a mixture will be uninterpretable.
2. As the reviewer makes it clear that he/she will be satisfied with some limited experimental interrogation of combinatorial effects, we used an alternative approach. We combined inactivation of Hoxb8 through glucocorticoid withdrawal (which is very efficient) with sgRNA mediated deletions of a TF (Section: "Double perturbations reveal TF interactions"). Specifically, we performed three combinatorial loss-of-function studies, by analysing the Cebpa/Hoxb8, Meis1/Hoxb8 and Spi1/Hoxb8 combinations, which highlighted complex patterns of regulation for hundreds of genes including many instances of dominant, buffering or synergistic effects. In addition, these new experiments validated our prediction that Hoxb8 is an upstream activator of Meis1, pinpointing the mechanisms of Hoxb8 operation as a key myeloid suppressor.

2. referee #2 states that the hierarchy between primary and secondary responses should be better addressed (ref#2, pt.2).

1. We fully agree with the reviewer that this is an important aspect to consider with any loss of function experiment. Unfortunately, genetic perturbation (CRISPR being the current method of choice) introduce a lag time, ranging widely for individual proteins (e.g. because of their different stability). Moreover, we would argue that the time points for the essential genes we are investigating are not late, but still relatively early, since it takes about 1 day for the virus to infect, reverse transcribe, integrate into the genome and start transcribing/translating (we know this because it takes about 24h for us to detect BFP expression following infection). This, combined with the lag time of protein decay, indicates that our 2-day measurements are still close to the acute timepoint of loss of the protein. Finally, there is a conceptual conundrum here with regards to TF knock-out experiments, which is that the "most important" target genes may well have the highest affinity binding sites in their enhancers/promoters, and therefore will remain expressed the longest while the amount of protein decays (this phenomenon has been described for example for Oct4 target genes in ES cells). Given all these considerations, we would argue that there is no optimal timepoint for gene expression analysis when using a CRISPR based approach, and the one that we used can be justified as well as several others. However, we thank the reviewer's comment as we now appreciate the insufficient justification in the original version, which has now been addressed in the revision (lines 134-137 and 175-182).

2. Following the reviewer's suggestion, we also explored in more depth the relationship between Gata3 and Ebf1 by using the ChIP-Seq and ATAC-Seq signatures as proxy for primary TF-target interaction. Lack of enrichment for genome-wide colocalisation indicates that Gata3 and Ebf1 do not bind regions in a coordinated manner. Nevertheless, Gata3 and Ebf1 may directly co-regulate a subset of their common targets (108 out of 475 genes) as we detected binding for both factors, albeit mostly at distinct locations within the gene loci. Taken together therefore, we conclude that the apparent correlation among Ebf1/Gata3 shared targets is most likely due to the direct Ebf1-Gata3 activation axis. We have commented on this issue in the revised version of the manuscript (lines 315-330).

Reviewers comments

Reviewer 1

In this study, Kucinsky et al. use of a fetal liver derived hematopoietic progenitor cell line in studying the role of 39 transcription factors with previously reported hematopoietic functions, using a knockout approach. Cells targeted for each of the individual factors were analyzed by single cell RNAseq. The results provided a complex picture of hematopoietic transcription factor networks active in cell fate decisions, some of which were known and others, such as the Ebf1-Gata3 synergism associated with the induction of a myeloid expression program, were unexpected and novel. The authors furthermore developed a new method (DoT) to visualize and predict cell fate transitions as a consequence of knocking out specific factors.

Besides delivering significant new insights into hematopoietic differentiation, the data generated in this study represent a valuable resource for investigators in the field and the DoT method should become a useful tool that can be applied for other types of single cell sequencing analyses. Overall the work is of high quality and the data well presented.

We thank the reviewer for highlighting the novelty of our findings and recognising the value of our work as a resource for the broader community. The reviewer also raises a number of specific points, which we have addressed as outlined below:

My only major problem with the paper is that it is very hard to read, which is in part based on the complexity of the problem and the computational analyses performed, some of which will not be familiar to most readers.

1. We were delighted in some ways that this was this reviewer's only major comment, but we certainly take it on board. We have asked a non-expert to read through the text, and point out any "dense"/hard to comprehend sections and modified them accordingly. We were also somewhat reassured by the following comment made by reviewer #3 (*"Overall, the MS is very well written and the results are described clearly"*).

1. Of the 39 genes that were perturbed, Rad21 is not known to encode a TF and this reviewer has not heard of noB. Please explain

1. The reviewer is correct that Rad21 has not been reported as a classical TF, but there are studies implicating it's binding to specific regions and transcriptional control in transcriptional control of pluripotent cells, e.g. <https://pubmed.ncbi.nlm.nih.gov/21589869/> Additionally, Rad21 mRNA levels are dynamic along major differentiation haematopoietic trajectories. We have added a statement to clarify this point (lines 144-148).

2. We apologize as this was a left-over short-hand label from draft versions of the figure. We have changed the noB to Hoxb8 in Figure 2A,B and 3B

2. Fig1C. What do the yellow and blue dots represent? Why are there only so few?

1. We have explained the projection in the figure legend. To reiterate the point also here, the projection score (based on nearest neighbours) reflects the relative transcriptional similarity to the Hoxb8-FL state for each LK/LSK cell. For the majority of cells no neighbours are identified (grey), some cells exhibit low similarity (yellow) and a small set of cells exhibit high similarity (blue).

3. FigS1F. The Venn diagram shows a surprisingly small degree of overlap between the 3 sgRNAs tested. Please comment.

1. While the overlap is not perfect, we wouldn't consider it small for an RNA-Seq experiment. Importantly, the example that we are showing here is middle of the range as far as all the TFs tested here are concerned. We felt that this was the fairest way of showing a representative example. The overlap for samples with strongest effects (Ebf1, Myc) is higher, and we provide additional examples below. It is also worth mentioning that the overlap between experiments and across different sgRNAs is predominantly dictated by the sensitivity threshold (lower sensitivity naturally leads to smaller overlap). Importantly, we observe strong positive correlations across the board, thus underlining the robustness of the datasets (8 replicates each for 3 different guide RNAs per gene).

4. Fig. 2B. This diagram is hard to understand. Specific interactions of factors depicted in the center (Myb, Cebpa, Runx1, etc.) are not well visualized. To improve this, additional representations of this graph would be useful to be shown in the supplement, such as blow-ups of some areas with the most relevant factors discussed later in the text. Also, the yellow circular symbols indicating perturbed TFs are very hard to see.

1. The figure provides a global view of the data to help illustrate the structure and complexity of the data, we realise due to the sheer size of the data not all the features are visible.
2. Per reviewer's suggestion we have now provided additional examples of specific regions in the Figure EV2.
3. We also increased the size of yellow dots, and increased the transparency of the edges to better expose the nodes.

5. The text does not specifically mention Figures 2A and B

1. We apologize for this oversight. Line 145 stated that “Figure 2 provides specific numbers of target genes and the network structure visualized as a force-directed layout”. We changed the wording to make it clear that this refers to Figure 2A,B

6. Fig. 3B and C. I find it very difficult to understand these figures. Why are only subsets of TFs represented in the two figures? Why does activation of a target in C not always show a positive correlation in B (e.g., Myc-Mitf)? Why are no downstream TF targets shown for e.g. C/EBPa, for which PU.1 has been described as a partner?

1. We apologize, as it seems that we didn't explain this well enough, we have improved the figure legend to describe this more clearly. The same set of TFs is used for all tiles in Figure 3, and the choice of these 'main' TFs was done for the sake of clarity (essentially, we excluded those TFs with very minor effects). We provide matching heat-maps for all TFs in Appendix Figure S3.
2. Some of the edges with small weights were filtered out for clarity. Full representation of that data can be read from Figure 3A.
3. With an activating link between two TFs (TF1 -> TF2), their overlapping targets indeed are expected to show positive correlation, but this may not always be the case due to: difference in timing (e.g. the secondary effects downstream of TF2 may not have appeared yet) or due to other regulatory characteristics. For instance, a feed forward loop (involving a third factor) could dampen the targets downstream of TF2. We have added relevant clarification in the main text (lines 175-182).
4. Please note that in the reviewer's example the link between Myc and Mitf is repressive rather than activating.
5. Figures 3A and B indeed both show positive correlation among targets shared by Cebpa and PU.1 (Spi1). Figure 3C does not show any TF downstream of Cebpa, but it is possible that some escaped our detection. Consistently with the reviewer's suggestions, PU.1 (Spi1) expression appears reduced after Cebpa knockout, but did not pass the threshold. We amended the text to clarify this important topic of thresholds (lines 152-155).

7. Fig.3D is another one that is hard to understand. The clusters selected do not seem to represent the major ones in the scheme. What are all the other clusters?

1. The modules represent groups of genes enriched for specific patterns of regulation by indicated TFs. We have modified the text to improve the explanation of the modules in the main text (lines 214-218).
2. We purposefully chose a range of modules to show their wide applicability and highlight the combinatorial nature of regulation. Some of the main (larger) modules are mostly regulated by a single TF which are less interesting from the module analysis standpoint albeit still very relevant for that particular regulator.
3. As to the nature of all the remaining modules, we provide all the regulated genes in Table EV4. This speaks to the resource aspect of the paper recognized by the reviewers, and will allow the wider research community to delve deeper into additional genesets and ontologies.

8. Line 219 '..we also generated chromatin immunoprecipitation (ChIP-Seq) datasets for 14 TFs as well as the H3K27Ac histone modification..' To judge the quality of the data please show representative examples in the Supplement

1. We have now provided representative UCSC views for all the ChIP-Seq data, and also a weblink to a UCSC genome browser session in the method sections, where readers of our paper can explore the status of their own favourite genes.

9. Line 291: 'Ebf1 counterbalances cell growth and cell cycle. How exactly are these distinguished?

1. Admittedly, we should have explained this aspect better. This statement is based on two pieces of evidence: opposing changes in expression of cell cycle genes and observed increase in cell growth simultaneous with a decrease in cell size. We have amended the relevant text to make this point clearer (lines 356-357).
2. Additionally, we provide, for the reviewer's evaluation graphs (below) representing expression changes of genes, whereby elevated expression is associated with specific cell cycle phases following Ebf1 knockout and Myc as a control. Without Ebf1, Hoxb8-FL cells upregulate preferentially the S, G2/M and M-phase genes, whereas lack of Myc leads to downregulation of G1/S and S-phase genes, consistent with our observations.

Figure - expression changes following Ebf1 knockout in Hoxb8-FL cells for genes highly expressed in specific cell cycle phases. Darker dots indicate genes classified as differentially expressed.

Figure - expression changes following Myc knockout in Hoxb8-FL cells for genes highly expressed in specific cell cycle phases. Darker dots indicate genes classified as differentially expressed.

10. Line 428: 'The network presented here also shows that Cebpa expression promotes the cell growth programme' (Figure 3D). How does this fit with the literature showing that C/EBPα inhibits growth?

1. The reviewer has highlighted an important aspect of anti-proliferative Cebpa function reported for myeloid progenitors. We suspect that this function may be cell-type specific. We find that Cebpa is essential for Hoxb8 cell growth, which is also reflected among Cebpa targets in our gene module analysis. This may be a combination of two factors: (1) Hoxb8 cells reflect more immature populations (than myeloid progenitors) and rely on balancing multiple lineage programmes to remain self-renewing and multipotent; (2) The Hoxb8-FL culture conditions contain only the Flt3L cytokine and no other cytokines associated with myeloid differentiation, possibly making cells more reliant on Cebpa for driving the myeloid programme. We have discussed this issue in the revised version of the manuscript (lines 520-523).

11. Fig.4. ChIPseq using what cells?

1. We thank the reviewer for spotting this omission, all ChIPseq experiments have been performed in Hoxb8-FL cells. We have amended the main text accordingly.

12. Fig. 5C. Please mention in the legend what is shown on the left and on the right.

1. We have improved the explanation of the 4 quadrants in the respective figure legend.

13. Show a reference image of the human single cell landscape with areas denoting different cell types.

1. Again, we should have stated this more clearly, because this information is already provided in Figure 7E. We added direct references to the annotation in the main text

Reviewer 2

Through the development of a combined CRISPR/Cas9 - RNAseq approach, the manuscript entitled "Cooperation of Lineage-associated Transcription Factors governs Haematopoietic Progenitor states" by Kucinski et al., aims to (a) investigate Transcription Factor (TF) -target gene dependencies, (b) the crosstalk between different TFs as well as (c) identify common regulatory pathways in hematopoietic progenitors. Furthermore, the authors developed a novel method (Direction of Transition: DoT) to estimate transcriptional changes in response to an insult (in this case KO of specific TFs).

HoxB8-FL cells, established in the Häkel lab and resembling LMPP in vitro, were chosen as a model system. Here, 39 different TFs, previously reported to be important in hematopoietic progenitors, were deleted using CRISPR/Cas9 technology. In order to address the functional connection between TF and target genes, only TFs causing the deregulation of more than 200 genes were subsequently followed up (19 out of 39). Integrating transcriptomic data after TFs KO, the authors were able to identify genes responding exclusively to a single TF, as well as genes dependent to two or more TFs. Furthermore, the authors identified also TF-TF dependencies elucidating not only crosstalk and co-regulations already known, but also novel interactions and novel roles (i.e. Ebf1). Finally, the authors used the plethora of data collected in this paper to develop a novel method to estimate phenotypical changes induced by specific alterations (in this case after TF KO).

The study is very stimulating and will help to shed light on the complex networks regulating hematopoietic progenitor states. The data and their analysis are of high quality. Furthermore, the study will be an inspiration for further research that use similar approaches to dissect regulatory networks along hematopoietic differentiation trajectories. The paper is very well written and it is easy to follow even though more technical and complex concepts are introduced. In general, the paper is highly interesting with significant novelty, which provides important new perspectives to the field.

We appreciate the positive feedback from the reviewer pointing out the newly identified interactions and praising the data quality and the clarity of the message. The reviewer also raised a number of specific points, which we have addressed as outlined below:

1. Despite the impressive amount of data collected and the deep analysis performed, functional validation experiments are missing. Furthermore, since single TFs are deleted one at the time and not simultaneously, the paper addresses neither cooperativity nor synergism between the different TFs, despite that fact that this is frequently claimed in the text (i.e. line 147, 153, 159, 175, 181). Indeed, the experimental design and the analysis shown in the paper can highlight correlation or anti-correlation of the transcriptional response after single TF KO, without the possibility to finely dissect the regulatory networks governing cell identity. Nevertheless, addressing cooperation or antagonism between the different TFs analyzed will be of great interest and will give additional strength to the current version of the paper. In this light, selected multiple TFs could be simultaneously deleted and the transcriptional response should be analyzed and compared to the changes induced by the deletion of each single TF.

1. We are grateful for this reviewer's comment, pointing out that synergy and cooperation convey specific meanings when considering the interaction between TFs. We have therefore changed our wording throughout the paper now, and use instead the terms 'co-regulation' or 'shared targets' as these do not imply a specific mechanism of action. To avoid any ambiguity, we rephrased relevant parts of the manuscript, specified what we mean by 'coregulation' and made it clear that we infer instances of co-regulation and anti-regulation among TF targets, based on the correlation and anti-correlation of their shared targets (e.g. lines 158, 169-171, 175-182).
2. We also fully recognise that experiments investigating TF cooperation through multiplex knock-out are of interest. However, we would also like to point out that in practical terms, this is not at all straightforward. Co-infecting cells with sgRNAs targeting two genes will generate a messy scenario, with approximately 50% of the cells not edited at all, 25% edited for just 1 gene, and only 25% being true double knock-outs. Carrying out RNA-Seq on such a mixture will be uninterpretable.
3. As the reviewer makes it clear that he/she will be satisfied with some limited experimental interrogation of combinatorial effects we used an alternative approach. We combined inactivation of Hoxb8 through glucocorticoid withdrawal (which is very efficient) with sgRNA mediated deletions of a TF (Section: "Double perturbations reveal TF interactions"). We performed these combinatorial perturbation assays for three combinations by analysing Cebpa/Hoxb8, Meis1/Hoxb8 and Spi1/Hoxb8. For many of the target genes, interactions were additive. However, these new experiments also highlighted complex patterns of regulation for hundreds of genes including many instances of dominant, buffering or synergistic effects. In addition, these new experiments validated our prediction that Hoxb8 is an upstream activator of Meis1, pinpointing the mechanisms of Hoxb8 operation as a key myeloid suppressor. These new results have been described in the new section: "Double perturbations reveal TF interactions".

2. Based on previous work from the lab, the authors divided the TFs in "essential" or "not essential". Based on this categorization, two different time points were chosen to perform transcriptomic analysis: 2 days after deletion for essential TFs, 4 days after deletion for non-essential TFs. Unfortunately, this experimental setting is sub-optimal since the time points chosen are relative late ones affecting the downstream interpretation of the data. Indeed, both primary and secondary targets will be identified in the analysis (as also stated in the text: line 231) with no possibility of distinguish between the two. Furthermore, after 2 days from the deletion of essential TFs (i.e. Myc), cells are already massively dying (as also shown in fig. S1G) adding an additional confounding factor to the analysis. This ambiguity could be the explanation of the massive transcriptional changes observed for some of the TFs, as well as for the numerous co-regulated targets identified. As also mentioned in the text, it is also a possible explanation for the poor correlation observed between transcriptional changes and binding of specific TFs to regulatory regions of deregulated genes. In order to distinguish between primary and secondary response and to give a more faithful screenshot of the functional interactions between TFs and target genes earlier time points should be included (at least for some of the candidates) and compared to the ones used in the current version of the manuscript.

1. We fully agree with the reviewer that this is an important aspect to consider with any loss of function experiment. Unfortunately, genetic perturbation (CRISPR being the current method of choice mid to high throughput studies) introduces a lag time, ranging widely for individual proteins (e.g. because of their different stability). Moreover, we would argue that the time points for the essential genes we are investigating are not late, but still relatively early, since it takes about 1 day for the virus to infect, reverse transcribe, integrate into the genome and start transcribing/translating (we know this because it takes about 24h for us to detect BFP expression following infection). This combined with the lag time of protein decay, indicates that our 2-day measurements are still close to the acute timepoint of loss of the protein. Finally, there is a conceptual conundrum here with regards to TF knock-out experiments, which is that the "most important" target genes may well have the highest affinity binding sites in their enhancers/promoters, and therefore will remain expressed the longest while the amount of protein decays (this phenomenon has been described for example for Oct4 target genes in ES cells). Given all these considerations, we would argue that there is no optimal timepoint for gene expression analysis when using a CRISPR based approach, and the one that we used can be justified as well as several others. However, we thank the reviewer's comment as we now appreciate the insufficient justification in the original version, which we have now addressed in the revision (lines 134-137 and 175-182).

3. The authors used CD45 locus as positive control to check deletion efficiency. While in the text it is claimed that 48% of the cells showed a successful CD45 inactivation, the FACS plot shown in Fig. S1A shows only 7% of the cells are BFP+ and have lost CD45 expression. The authors should explain this discrepancy.

1. We are sorry for the confusion here – clearly we didn't explain this well enough in the figure/figure legends. The 7% value is a fraction of total cells analysed. Editing efficiency is calculated out of the 14.63% of successfully infected cells, as these are the only ones expressing sgRNA. We have annotated the figure and the legend to make this more clear.

4. Even though the efficiency of targeting was proven using CD45 as positive control, as well as showing the fraction of genomic DNA reads with frameshifts for a selected number of TFs, either gene expression or protein analysis should be provided by the authors, at least for some prominent examples, to demonstrate efficient reduction of mRNA and/or protein of all the TFs used in the study.

1. We appreciate the reviewer's suggestion. Unfortunately, as we are relying on indel mutations introduced by Crispr/Cas9, these may not always lead to mRNA depletion (some mRNAs are expected to be degraded by nonsense-mediated decay). Consistently, in most cases we observed reduced expression levels for targeted TFs.
2. To provide further evidence that our Crispr/Cas9 approach produces loss-of-function mutations we attempted to isolate single cell clones of Hoxb8-FL cells following sgRNA treatment. We managed to isolate clones for Hoxb8-FL cells lacking Gata3, despite their poor survival. The ChIP-Seq experiment using Gata3 antibody performed in these setting only shows background signal akin to an IgG control ChIP-Seq experiment (UCSC browser session: http://genome-euro.ucsc.edu/s/idk25/TFnet2020_allChIPs_impr), confirming the lack of functional Gata3 protein as well as the antibody specificity (lines 121-124).

5. The authors suggest a novel mechanism of action for Ebf1 working in concert with Gata3. Nonetheless, due to the failure to discriminate between primary and secondary response, and since the authors showed that Gata3 is actually an Ebf1 target, it is reasonable to hypothesize that most of the shared target genes are mostly Gata3 ones. In order to address this point, transcriptomic analyses using shorter time point, as well as integration analysis between RNAseq and ChIPseq should be performed in order to identify Ebf1 unique targets along with Ebf1/Gata3 shared ones.

1. Following the reviewer's suggestion we explored more in depth the Gata3 and Ebf1 relation by using the ChIP-Seq and ATAC-Seq signatures as proxy for primary TF-target interaction. There is no statistically significant enrichment for colocalisation thus indicating that Gata3 and Ebf1 do not bind regions in a coordinated manner. Gata3 and Ebf1 may directly co-regulate a subset of their targets (108 out of 475 genes) as we detected binding for both factors by ChIP, which however tended to be at distinct locations within the given gene loci. Taken together therefore, we would conclude that the apparent correlation among Ebf1/Gata3 shared targets is most likely due to the Ebf1-Gata3 activation axis. We have commented on this issue in the revised version of the manuscript (lines 315-330).

6. In order to identify groups of genes similarly deregulated by different TFs, the authors subdivide the transcriptomic changes into 47 modules. In the text, few key genes belonging to crucial molecular pathways were mentioned, yet, a global pathway enrichment analysis is missing. In order to have a clear overview of genes contained in each module, authors should perform pathway enrichment analysis (IPA or others).

1. We now provide gene ontology enrichment tables for the relevant modules.

7. Supplementary tables provided are not correct: they show data not related to the manuscript submitted.

1. From what we can see from our end of the online submission portal, the uploaded tables are correct. If this problem persists, we hope that the editor will be able to help out.

8. Network analysis shown in Fig.3C does not highlight CEBP β regulation mediated by Spi1, as instead mentioned in the text (line 173). Please elaborate better or specify in the text which is the corresponding figure.

1. This is a mistake on our part. We have removed Spi1 from the text.

9. Often, along the text, deregulation of different genes is mentioned (i.e. line 175, 176, 193, among others). Please provide plots where expression levels of the mentioned genes are shown.

1. We have provided respective plots with gene expression changes for examples of deregulated genes throughout the text (Appendix Figure S2).

Reviewer 3

The MS presents new data on the genetic basis of hematopoietic progenitor cell differentiation obtained by systematic KO of 39 TFs in a cell line model for multipotent progenitors (Hoxb8-FL). Data from ChIP-seq and single-cell RNA-seq experiments of these cells and ex vivo isolated hematopoietic stem and progenitor cells are integrated with this new data set. Also, a bioinformatic method, Direction of Transition, is introduced that aims at predicting how cell shift in the transcriptional landscape upon TF knockdown (or any other type of perturbation).

The new data obtained upon TF KO are a very important resource for the field, as the authors already demonstrate with several new mechanistic insights. These include the co-expression of Ebf1 and Gata3 (mediated at least in part by Ebf1 activating Gata3 transcription) in lymphoid-primed progenitors prior to committing to B cell development.

Overall, the MS is very well written and the results are described clearly.

We are very grateful for the positive comments highlighting new mechanistic insights and the quality of the manuscript. The reviewer also raises a number of specific points, which we have addressed as outlined below:

1. Direct versus indirect TF effects (p.6: "... much of this uncertainty can be disentangled"): Could the authors be more specific here? Does knowing the network of TF interactions help decide whether the KO effect of one of those TFs on another target gene is direct or indirect? Can this be quantified?

1. We appreciate the reviewer's comment, which made us realise that we did not provide sufficient explanation for the parallel/hierarchical interactions among TFs. We modified the results section to illustrate these concepts and explicitly state that while it's difficult to pinpoint primary targets, hierarchical regulation can be inferred directly from our network (lines 175-182). This helps interpreting instances of apparent overlap and correlations of target genes.
2. Suspected primary interactions can also be cross-referenced with our ChIP-Seq data. We have added a link to an interactive UCSC session to facilitate such analysis for the potential reader.
3. Quantification suggested by the reviewer would be extremely useful but is complicated by two confounding factors: unequal TF knockout efficiency across conditions and the unknown timing of TF protein decay (thus induced target expression changes) following the knockout. We believe that future advances in CRISPR/Cas9 technology and time-resolved data will unlock this exciting prospect.

2. Fig. 3C: Reduce font size so as not to obscure arrowheads in a few places

1. We have reduced the font size in Figure 3C and shifted the labels so that they do not obscure the arrows.

3. Fig. 4C,D: Improve annotation. Where is the +37kB enhancer for Cebpa in C? What does the red arrow point to in D?

1. We have annotated the Cebpa enhancer in Figure 3C and added explanation of the red arrow in the figure legends, which points to a putative enhancer element downstream of Gata3.

4. Fig. 6: Please explain A in some detail (in text or figure legend). Is there a viewpoint region in the landscape?

1. We appreciate highlighting the insufficient explanation. We now provide a step-by-step explanation in the legend of Figure 6, a visual representation of vectors and highlighted the viewpoint in Figure 6. For clarity, we chose a cluster of intermediate progenitors as the viewpoint.
2. **It is not clear whether Euclidian distance is the right metric. If the transcriptional landscape is a nonlinear manifold in gene-expression space, then diffusion distance may be more appropriate. Please discuss.**
 1. The reviewer is correct that the vectors used for DoT score computation do not follow the manifold. However, using the diffusion distance requires partitioning the cells into defined trajectories. This is an actively developing area of research without an established consensus. Nevertheless, the DoT score method can be easily combined with flexible and accurate method (Saelens et al. 2019) - PAGA (Wolf et al. 2019). By considering DoT scores in cells belonging to the adjacent clusters identified with graph abstraction, we can highlight directions in the naturally occurring paths of differentiation. We provide an example visualisation below, PAGA graph is colour-coded based on the DoT scores, cluster 4 is the viewpoint.
 2. We modified the text to make this aspect of DoT score more explicit and included a recommendation for using trajectory inference methods to aid the interpretation (lines 412-416).
 3. Using vectors in gene expression space carries also an advantage in case of discontinuous landscapes. Please consider, for instance, the Ikzf1 and T cells signature. DoT score helps annotating genes downstream of Ikzf1 relevant to the T cell state (which is disconnected from the main landscape).

6. How does the direction vector field with respect to a viewpoint compare with RNA velocity?

1. The reviewer draws an interesting parallel between the RNA velocity and DoT scores, which both help to predict (or interpret) a cell state. DoT score uses perturbation/treatment data, hence is compatible with various single-cell or bulk expression data. RNA velocity attempts to predict future cell state based on the spliced/un-spliced RNA information.

2. RNA velocity operates on much shorter time-scales (few hours), our data may not be the best system for direct comparison as it analyses longer time-points. Additionally, we analyse small bulk populations, which are not directly part of the reference landscape (though correspond to a relatively well-defined group).
3. Nevertheless, the RNA velocity should be compatible with nascent RNAs identified with e.g. scSLAM-Seq (Erhard et al. 2020, Battich et al. 2020). These methods are likely to prove useful in the future for highlighting immediate (within a few hours) changes in expression. In such settings, the DoT-score method can be employed to interpret expression changes in complex landscapes. We are thankful for raising this issue, which we now discuss in the revised manuscript (lines 456-460).

Dear Bertie,

Thank you for submitting your amended manuscript (EMBOJ-2020-104983R) to The EMBO Journal. My apologies for the delay with the re-review of your manuscript at this time of the year due to protracted referee input. Your revised study was sent back to referees #1 and #2 for re-evaluation, and we have received comments from referee #1, which I enclose below. This referee finds that his-her concerns have been sufficiently addressed and s/he is now broadly in favour of publication. Please note that while referee #2 was at this time not able to look back into the revised study, we have editorially evaluated your response to the concerns raised by both referees #1 and #3 and found them to be reasonably considered.

Thus, we are pleased to inform you that your manuscript has been accepted in principle for publication in The EMBO Journal, pending some minor remaining issues related to formatting and data representation as detailed below which need to be addressed at re-submission.

Please contact me at any time if you have further questions.

Thank you for giving us the chance to consider your manuscript for The EMBO Journal. I look forward to your final revision.

Again, please contact me at any time if you need any additional help.

Best regards,

Daniel

Daniel Klimmeck PhD
Editor
The EMBO Journal

>> Please add a separate 'Statistical analysis' section in the material & methods part.

>>Rename the current 'Authorship' section to 'Author contributions'. Specify contributions for R.H. and S.K. .

>> Re-check callouts for 'Figure EV4' in the manuscript. Add callouts for Tables EV1 and EV2.

>> We do not offer the option to state 'Data not shown' as to our policies. Please either add respective data sets or remove statements at pages 33 and 37.

>> Please add a ToC on the first page of the appendix.

>> Adjust the reference format limiting to 10 authors et al. .

>> Dataset EV legends: please remove table legends from manuscript and add them directly to the respective files. All files should be renamed "Dataset EV1" except for Table EV 2, which would consequently need to be renamed Table EV1.

>>Please enter the funding information (grant numbers) to your manuscript to match the information provided in our online manuscript system.

>> Remove GEO web links from the data availability section.

>> Add a separate statistical analysis paragraph to the material and methods part of the manuscript.

>> Please consider additional changes and comments from our production team as indicated by the .doc file enclosed and leave changes in track mode.

- a point-by-point response to the referees' comments, with a detailed description of the changes made (as a word file).

- a word file of the manuscript text.

- individual production quality figure files (one file per figure)

- a complete author checklist, which you can download from our author guidelines

(<https://www.embopress.org/page/journal/14602075/authorguide>).

- Expanded View files (replacing Supplementary Information)

Further information is available in our Guide For Authors:

The revision must be submitted online within 90 days; please click on the link below to submit the revision online before 30th Nov 2020.

Link Not Available

Referee #1:

I am happy with the responses made by the authors, except for criticism 9) for which the authors now provided two graphs for cell cycle specific expression of genes in Ebf1 and Myc KO cells. For Ebf1 they conclude 'Without Ebf1, Hoxb8-FL cells upregulate preferentially the S, G2/M and M-phase genes,.. However, to this reviewer the only obvious change is an increase in G2/M genes. This suggests that Ebf1, in contrast to Myc, facilitates the transition into the G2/M phase. Therefore the authors might want to slightly modify the interpretation of their data.

The authors performed the requested editorial changes.

Dear Bertie,

Thank you for submitting the revised version of your manuscript. I have now evaluated your amended manuscript and concluded that the remaining minor concerns have been sufficiently addressed.

Thus, I am pleased to inform you that your manuscript has been accepted for publication in the EMBO Journal.

Please note that it is EMBO Journal policy for the transcript of the editorial process (containing referee reports and your response letter) to be published as an online supplement to each paper. I would thus like to ask for your consent on keeping the additional referee figures included in this file.

Also in case you might NOT want the transparent process file published at all, you will also need to inform us via email immediately. More information is available here:

http://emboj.embopress.org/about#Transparent_Process

Please note that in order to be able to start the production process, our publisher will need and contact you regarding the following forms:

- PAGE CHARGE AUTHORISATION (For Articles and Resources)

[http://onlinelibrary.wiley.com/journal/10.1002/\(ISSN\)1460-2075/homepage/tej_apc.pdf](http://onlinelibrary.wiley.com/journal/10.1002/(ISSN)1460-2075/homepage/tej_apc.pdf)

- LICENCE TO PUBLISH (for non-Open Access)

Your article cannot be published until the publisher has received the appropriate signed license agreement. Once your article has been received by Wiley for production you will receive an email from Wiley's Author Services system, which will ask you to log in and will present them with the appropriate license for completion.

- LICENCE TO PUBLISH for OPEN ACCESS papers

Authors of accepted peer-reviewed original research articles may choose to pay a fee in order for their published article to be made freely accessible to all online immediately upon publication. The EMBO Open fee is fixed at \$5,200 (+ VAT where applicable).

We offer two licenses for Open Access papers, CC-BY and CC-BY-NC-ND.

For more information on these licenses, please visit: <http://creativecommons.org/licenses/by/3.0/> and http://creativecommons.org/licenses/by-nc-nd/3.0/deed.en_US

- PAYMENT FOR OPEN ACCESS papers

You also need to complete our payment system for Open Access articles. Please follow this link and select EMBO Journal from the drop down list and then complete the payment process:

https://authorservices.wiley.com/bauthor/onlineopen_order.asp

Should you be planning a Press Release on your article, please get in contact with embojournal@wiley.com as early as possible, in order to coordinate publication and release dates.

On a different note, I would like to alert you that EMBO Press is currently developing a new format for a video-synopsis of work published with us, which essentially is a short, author-generated film explaining the core findings in hand drawings, and, as we believe, can be very useful to increase visibility of the work.

Please see the following link for a representative example:

http://embopress.org/video_EMBOJ-2014-90147

If you have any questions, please do not hesitate to call or email the Editorial Office.

Best regards,

Daniel

Daniel Klimmeck, PhD
Editor
The EMBO Journal
EMBO
Postfach 1022-40
Meyerohofstrasse 1
D-69117 Heidelberg
contact@embojournal.org
Submit at: <http://emboj.msubmit.net>

Corresponding Author Name: Gottgens, Berthold

Manuscript Number: EMBOJ-2020-104347